# Convergent allostery in ribonucleotide reductase

William C. Thomas[1,2], F. Phil Brooks III[2], Audrey A. Burnim [1,2], John-Paul Bacik[1,2], JoAnne Stubbe[3], Jason T. Kaelber [4], James Z. Chen[5] & Nozomi Ando[1,2]

Ribonucleotide reductases (RNRs) use a conserved radical-based mechanism to catalyze the conversion of ribonucleotides to deoxyribonucleotides. Within the RNR family, class Ib RNRs are notable for being largely restricted to bacteria, including many pathogens, and for lacking an evolutionarily mobile ATP-cone domain that allosterically controls overall activity. In this study, we report the emergence of a distinct and unexpected mechanism of activity regulation in the sole RNR of the model organism *Bacillus subtilis*. Using a hypothesis-driven structural approach that combines the strengths of small-angle X-ray scattering (SAXS), crystallography, and cryo-electron microscopy (cryo-EM), we describe the reversible interconversion of six unique structures, including a flexible active tetramer and two inhibited helical filaments. These structures reveal the conformational gymnastics necessary for RNR activity and the molecular basis for its control via an evolutionarily convergent form of allostery.

[1] Department of Chemistry and Chemical Biology, Cornell University, Ithaca, NY 14853, USA. [2] Department of Chemistry, Princeton University, Princeton, NJ 08544, USA. [3] Department of Chemistry, Massachusetts Institute of Technology, Cambridge, MA 02139, USA. [4] Institute for Quantitative Biomedicine, Rutgers University, Piscataway, NJ 08854, USA. [5] Department of Biochemistry and Molecular Biology, Oregon Health & Science University, Portland, OR 97239, USA. Correspondence and requests for materials should be addressed to N.A. (email: nozomi.ando@cornell.edu)

Ribonucleotide reductases (RNRs) catalyze the conversion of ribonucleotides to deoxyribonucleotides, a conserved reaction that is fundamental to DNA-based life. As essential enzymes in central metabolism[1,2], RNRs have evolved two complex forms of allostery[3–5]: specificity regulation, which is conserved in all RNRs, and activity regulation, which is canonically attributed only to RNRs with an evolutionarily mobile, regulatory domain known as the ATP-cone[6]. In this study, we describe the unexpected emergence of a convergent form of activity regulation in the class Ib RNRs, a major subset of aerobic RNRs that lack ATP-cones[7–11]. The evolution of class Ib RNRs is particularly relevant to medicine, as they are the primary aerobic RNRs used by a number of bacterial pathogens, such as *Bacillus anthracis*, *Mycobacterium tuberculosis*, *Staphylococcus aureus*, and *Streptococcus pneumoniae*[12].

Nearly all aerobic organisms use class I RNRs, which consist of two subunits: α, which contains the catalytic site (Fig. 1a), and β, which houses a subclass-dependent metal center (Fig. 1b). Enzyme activity requires the coordination of three processes: radical generation, nucleotide reduction, and re-reduction of the catalytic site. The first step involves reversible, long-range (~35 Å) radical transfer (RT) from a stable tyrosyl radical cofactor in β to a central cysteine in the catalytic site of α (Supplementary Fig. 1a). The flexible C-terminus of β plays a critical role in this process by binding to α and contributing a conserved tyrosine along the RT pathway[13–16] (Supplementary Fig. 1c). Once the thiyl radical is generated in α, nucleotide reduction proceeds via a conserved mechanism using two additional redox-active cysteines in the catalytic site as reducing equivalents[2]. The resulting disulfide is then reduced by a cysteine pair on the flexible C-terminus of α[17] (Supplementary Fig. 1d). Because RT is required for turnover, the active class I RNR complex is generally thought to adopt a compact conformation similar to a previously proposed symmetric $\alpha_2\beta_2$ docking model[18] (Supplementary Fig. 1a). However, the structural basis for RNR activity has not yet been elucidated at high resolution.

Allosteric regulation of RNRs plays a key role in maintaining the appropriate balance of intracellular nucleotides required for DNA replication fidelity. Class I RNRs reduce ribonucleoside diphosphates (NDPs, where N is any of the four bases, A, U, C, or G), with additional enzymatic steps converting the dNDP products into dNTPs. Substrate preference is allosterically coupled to the binding of a dNTP in the specificity or S-site (Fig. 1a), thus ensuring the balance of dNTP pools[3,19]. In class I RNRs, the S-site is located at the interface of a canonical $\alpha_2$ dimer, hereafter denoted the "S-dimer" (Fig. 1a). Additionally, many RNRs from every class are able to regulate overall activity via a second allosteric site designated the activity or A-site (Fig. 1d). This site is housed in an evolutionarily mobile ~100-residue ATP-cone domain composed of a four-helix bundle and a three-stranded β-sheet cap, which is typically found at the N-terminus of the α subunit[6,11,20] (Fig. 1d, f). Binding of the substrate derivative ATP increases activity, while the downstream product, dATP, acts as a feedback inhibitor.

Most class I RNRs are classified into two subgroups on the basis of the radical-generating cofactor. Class Ia RNRs (NrdA = α, NrdB = β) are found in all domains of life and utilize a diferric tyrosyl cofactor. In contrast, class Ib RNRs (NrdE = α, NrdF = β) employ a dimanganic tyrosyl cofactor[21–23] and are thought to have evolved from a Ia progenitor as a secondary aerobic RNR in bacteria[24]. Studies of class Ia RNRs have shown that ATP-cone domains preferentially interface with other RNR chains (α or β) upon binding of dATP to form ring-shaped oligomers that are proposed to inhibit activity by preventing long-range RT[5,25–28]. Whereas most class Ia RNRs have at least one ATP-cone, class Ib RNRs are notable for sharing a characteristic truncated ATP-cone

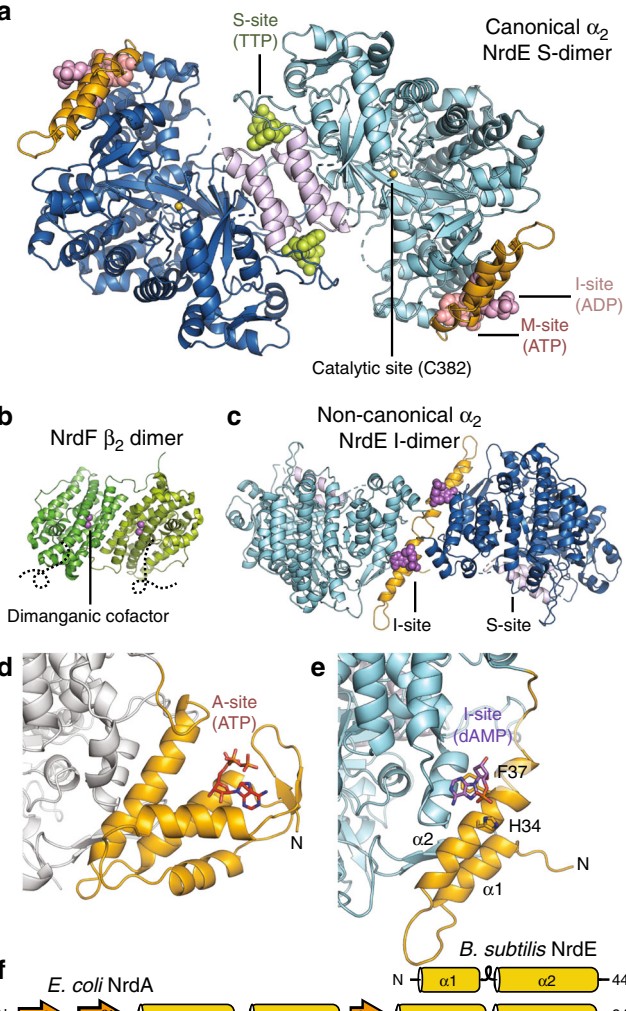

**Fig. 1** Allosteric sites of the *Bacillus subtilis* class Ib ribonucleotide reductase. **a** A 2.50 Å crystal structure of *B. subtilis* NrdE (α subunit) obtained under activating conditions depicts an S-shaped dimer ("S-dimer") interfacing at the "specificity" or S-site (lavender). A specificity effector TTP (green) is bound to the S-site, and activating nucleotides, ADP (pink) and ATP (salmon), are bound to two allosteric sites that evolved near the N-terminus of *B. subtilis* NrdE. A catalytically essential radical is generated at a central cysteine in the catalytic site, C382 (yellow sphere). **b** *B. subtilis* NrdF (β subunit) is dimeric and utilizes a dimanganic tyrosyl cofactor (purple spheres) to initiate radical chemistry (PDB: 4DR0)[21]. A disordered region of the NrdF C-terminus (black dotted lines) is critical for radical transfer. **c** A recent structure of *B. subtilis* NrdE co-crystallized with dAMP (purple) depicts a partially inhibited, non-canonical "I-dimer" with the interface formed by the truncated ATP-cone (orange) (PDB: 6CGL)[31]. **d** In class Ia RNRs, ATP or dATP binds to the "activity" or A-site in the ATP-cone domain (orange) to mediate changes in quaternary structure and tune overall activity (PDB: 3R1R)[20]. **e** Class Ib RNRs only contain partial ATP-cones (orange). *B. subtilis* NrdE is unusual in that it displays activity regulation and binds dAMP (purple) in the N-terminally located I-site (PDB: 6CGL). **f** The partial N-terminal cone of class Ib RNRs (top) is structurally homologous to the last two helices of the canonical ATP-cone found in many class Ia RNRs (bottom) but lacks A-site residues

sequence at the N-terminus of α (NrdE) and for lacking key A-site residues[7,11,29] (Fig. 1e, f, orange). Consistent with this, class Ib RNRs have generally shown a lack of activity regulation[7–10].

Contrary to all other class Ib RNRs studied to-date, the sole RNR of *Bacillus subtilis* displays class Ia-like activity regulation[30].

Recently, we found that dATP inhibition in *B. subtilis* RNR is enhanced by the presence of a nucleotide with no previously documented role in RNR regulation: an endogenous deoxyadenosine monophosphate (dAMP) that co-purifies with NrdE[31]. A co-crystal structure revealed dAMP bound to an unexpected site in the truncated N-terminal cone domain to promote α dimerization at a non-canonical interface (Fig. 1c, PDB: 6CGL) rather than the expected S-dimer interface. We denote this dAMP-binding site the "I-site" (Fig. 1e) for its potential role in inhibition and correspondingly denote this structure the "I-dimer". Although the structural basis for class Ia-like behavior was not elucidated, the new ligand and dimer implied a form of inhibition not yet seen in other RNRs[31].

Motivated by this discovery, we applied a structural approach driven by small-angle X-ray scattering (SAXS) to elucidate the molecular basis for activity regulation in *B. subtilis* RNR. Because unique, thermodynamically stable states can be mathematically distinguished by SAXS[5], we first used this technique to map the reversible and dynamic interconversion of six distinct oligomerization states in solution, including that of a flexible active complex and inhibited helical filaments (Supplementary Tables 1 and 2). Drawing on the conformational landscape established with SAXS, structures of individual states were systematically determined by cryo-electron microscopy (cryo-EM) and crystallography (Supplementary Tables 3–5). We find that *B. subtilis* RNR has arrived at a remarkably similar solution to the ATP-cone through the evolution of two unique allosteric sites that modulate catalytic activity by inducing contrasting quaternary interactions that either enable or prevent the full range of motions needed for RNR activity.

## Results

**dATP binds to two allosteric sites to form a filament.** To examine the role of the unusual dAMP ligand in activity regulation, studies were conducted on both the dAMP-free and dAMP-bound species, hereafter referred to as "apo" and "holo" to describe the initial ligand state. Two types of SAXS experiments were conducted in this study (Supplementary Table 1). First, nucleotide and subunit titrations were performed to characterize structural transitions. Then, to obtain conformationally pure scattering profiles, SAXS was performed with in-line size-exclusion chromatography (SEC–SAXS), and the resultant datasets were mathematically decomposed with evolving factor analysis (EFA), a method we developed to separate dynamically exchanging species that partially co-elute[32] (Supplementary Table 2).

We first confirmed with SEC–SAXS that in the absence of dATP, apo-NrdE is largely monomeric (Fig. 2a, blue), having a radius of gyration $R_g$ of $29.5 \pm 0.1$ Å, whereas holo-NrdE is predominantly I-dimer, having an $R_g$ of $44.0 \pm 0.1$ Å (Fig. 2a, orange). A systematic titration of dATP was then performed on both forms of NrdE at a physiologically relevant concentration of 4 μM α with a saturating substrate concentration of 1 mM CDP. For both forms of NrdE, titration of dATP leads to a sharp increase in $R_g$, approaching ~85 Å at inhibiting levels of dATP (50 μM). The $R_g$ curve of apo-NrdE (Fig. 2b, gray) lags behind that of holo-NrdE (Fig. 2b, orange), suggesting that dAMP and dATP have partially overlapping effects. Singular value decomposition (SVD) yields two significant singular values for each individual dataset, whereas SVD of the two datasets combined yields three (Supplementary Fig. 2a). Because the monomer and I-dimer account for two singular values, the shared third singular value indicates that the final state is the same for both forms of NrdE.

To structurally characterize the dATP-induced oligomer, SEC–SAXS was performed on holo-NrdE with 100 μM dATP

and 0.5 mM CDP in the running buffer (Supplementary Fig. 2b, top). The scattering curve of the predominant scattering component features a prominent secondary peak at $q \sim 0.077$ Å$^{-1}$ (Fig. 2a, red curve and star), suggestive of an extended oligomer with a hollow, cylindrical shape (Supplementary Fig. 2b, bottom). Together, these SAXS results indicate that the I-dimer is a component of a filament that forms in the presence of dATP. However, since the formation of this oligomer can occur without dAMP initially present, dATP is likely to be able to perform dAMP's role, i.e. binding the I-site, in addition to binding another site.

**Cryo-EM structures of dATP-inhibited filaments.** Having obtained SAXS-based evidence for dATP-induced filament formation, structure determination was undertaken with cryo-EM (Supplementary Table 3, Supplementary Figs. 3–5). A 6 Å resolution cryo-EM map was obtained of holo-NrdE under inhibiting conditions (100 μM dATP, 1 mM CDP), revealing an unusual double-helix with a hollow center (Fig. 2d). In this structure, each helical strand is composed of alternating I-dimer and S-dimer interfaces, leading to a non-terminating chain of NrdE subunits (Fig. 2d). Importantly, the simulated scattering from this double-helical model captures the distinct secondary scattering peak observed in the experimental scattering (Fig. 2a, black dashed), indicating that this structure is a stable species in solution.

Because enzyme activity requires both α and β subunits, cryo-EM was repeated with mixtures of holo-NrdE and NrdF (abbreviated as "NrdEF") under inhibiting conditions. The resultant cryo-EM map revealed a NrdEF filament consisting of a single-helical NrdE strand (Fig. 2e, gray surface) with EM density corresponding to NrdF observed at higher contour levels filling the helical interior (Fig. 2e, green). The resolution of the reconstruction varied across the map, as high-resolution features were distinguishable in NrdE but not in NrdF (Fig. 2e, Supplementary Fig. 5b, d). Additional density was observed in a hydrophobic cleft of NrdE (Fig. 2e, f, green) that was previously shown to bind the β C-terminus in other class I RNRs. Using a previous structure[33] as a guide, we were able to model the bound NrdF C-terminus as an eight-residue polyalanine chain (Fig. 2e, f, see the section "Methods"). In the NrdE-only filament, this cleft is unoccupied and partially buried by the double-helical interface formed by residues 660–665 (Supplementary Fig. 6a). Thus, the dissociation of the double-helical NrdE filament is likely a result of the NrdF C-terminus competing with this interface. These observations are consistent with a subunit titration performed with SAXS, in which addition of NrdF to the NrdE filament led to the immediate disappearance of the secondary peak (Fig. 2c, red star; Supplementary Fig. 4d).

For both filaments, excess density at the S-site was modeled as dATP (Fig. 2g). As dATP acts as a specificity effector for CDP reduction, its presence at this site explains the formation of the S-dimer interface between adjacent I-dimer units. Additionally, a distinct, protruding density is observed at the I-site that is best explained as a dATP that displaced dAMP (Fig. 2h). Side-chain densities in the I-dimer interface (Supplementary Fig. 6c) are modeled well by a previously reported crystal structure of the dAMP-bound I-dimer (PDB: 6CGL)[31]. A key feature of this interface is the "F47-loop" comprising residues 45–50, which immediately follows the partial ATP-cone sequence (Fig. 1f). The F47-loops from adjacent monomers interlock at the I-dimer interface, with F47 and H49 reaching across to interact with the opposing chain (Supplementary Fig. 6c). Our structures indicate that unlike dAMP, dATP can bind both allosteric sites, which further implies that any combination of a deoxyadenosine nucleotide at the I-site and a specificity effector at the S-site

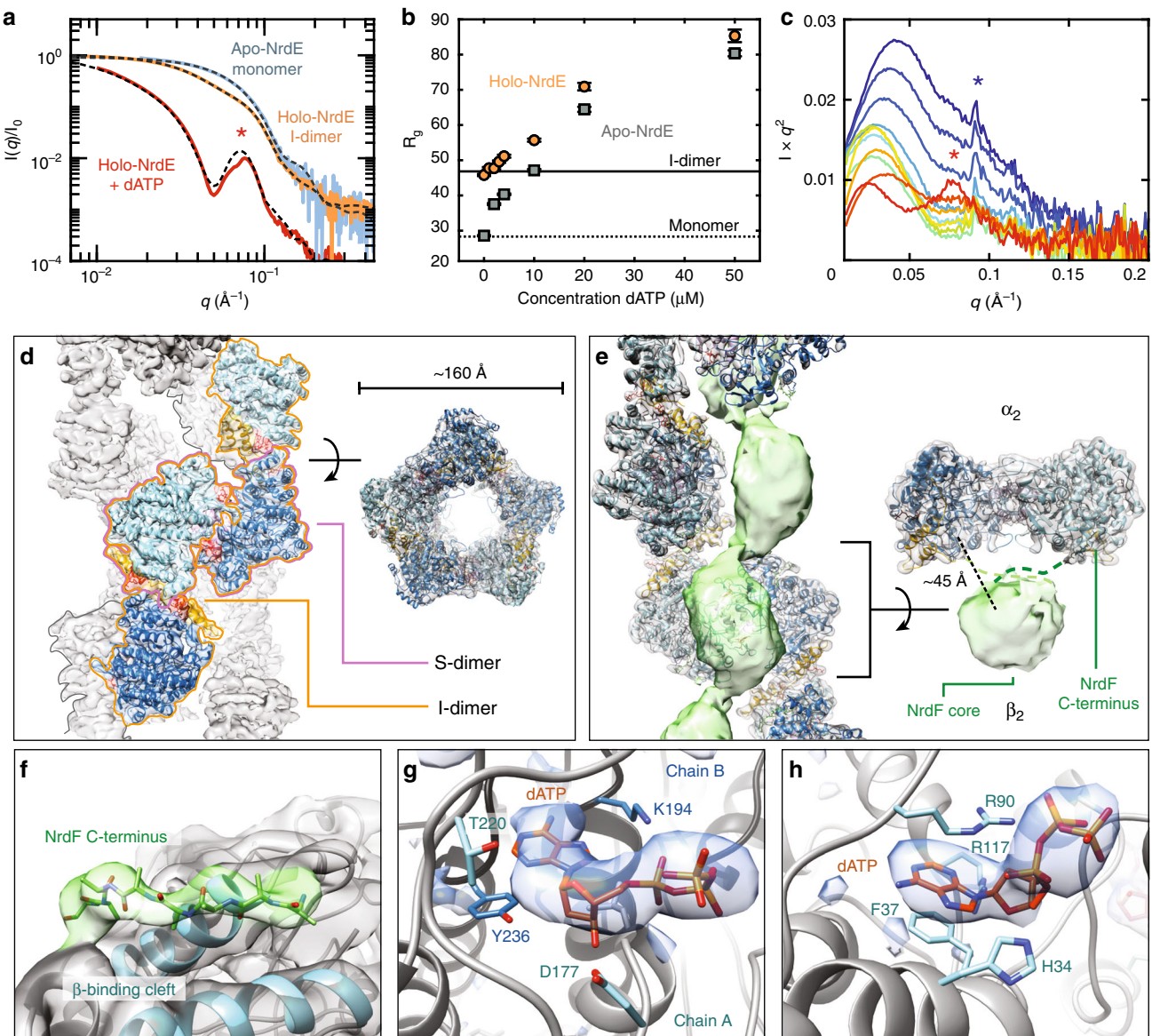

**Fig. 2** The inhibitor dATP stabilizes an unusual helical NrdE filament. **a** EFA-separated SEC–SAXS profiles of apo-NrdE (blue), holo-NrdE (orange), and holo-NrdE + 100 μM dATP (red) agree with the theoretical scattering (dashed) of a monomer[31], I-dimer[31], and a 34-mer double-helix model derived from cryo-EM. **b** SAXS titrations of 0–50 μM dATP to 4 μM holo-NrdE (orange) and apo-NrdE (gray) display increasingly larger $R_g$ values, suggestive of non-terminating oligomerization. Theoretical $R_g$ values of the I-dimer (46 Å, solid line) and monomer (27 Å, dotted line) are shown for comparison. **c** A SAXS titration of 0–20 μM NrdF to 4 μM C382S holo-NrdE + 50 μM dATP (red to blue), plotted in Kratky representation, shows the loss of the NrdE double-helix (red star) with the formation of the NrdEF filament. This change is followed by an accumulation of excess NrdF and eventual appearance of ordered NrdEF filament assemblies (blue star, Supplementary Fig. 4b, d). **d** A 6 Å cryo-EM map of the dATP-induced NrdE filament (threshold = 4.45) reveals a double-helical structure with each helix formed by alternating S-dimer (pink outline) and I-dimer (orange outline) interfaces. **e** A 4.7 Å cryo-EM map of dATP-inhibited NrdEF displays strong density for a helical NrdE filament (gray, threshold = 1.14) and weaker density for NrdF forming a central column of beads (green, threshold = 0.45). Each ASU consists of a NrdF dimer centered on a NrdE S-dimer, resulting in an $\alpha_2\beta_2$ with a prominent gap (right). **f** The NrdF C-terminus (green sticks/surface, threshold = 1.18) is observed at high occupancy in the β-tail-binding cleft of NrdE (blue cartoon, gray surface, threshold = 1.18). **g** and **h** Difference densities in the NrdEF cryo-EM map for (**g**) dATP in the S-site (threshold = 7.73) and (**h**) dATP in the I-site (threshold = 12.6). Corresponding σ levels are estimated in Supplementary Table 5. Source data are provided as a Source Data file

would induce dimerization at both interfaces and subsequent filament formation.

Density for the NrdF core is significantly weaker than the remainder of the NrdEF map (Supplementary Fig. 5b, d), but density for the NrdF C-terminus is observed at similar thresholds as NrdE (Fig. 2f). This discrepancy suggests that the NrdF core is flexibly linked to NrdE by NrdF C-termini, resulting in high occupancy but significant variability in orientation. The NrdF core volume is of the appropriate size and shape to be best

explained by a NrdF dimer symmetrically centered with respect to a NrdE S-dimer, resulting in an $\alpha_2\beta_2$ asymmetric unit (ASU) (Fig. 2e, right). Attempts to refine the map from biased initial models with little gap between NrdE and NrdF invariably converged to have a pronounced gap of roughly 15–20 Å between the subunits (see the section "Methods"). Thus, instead of forming a buried interface with NrdE, the NrdF core density is on average placed far from the catalytic sites. Such a gap at the subunit interface would increase the distance between the buried

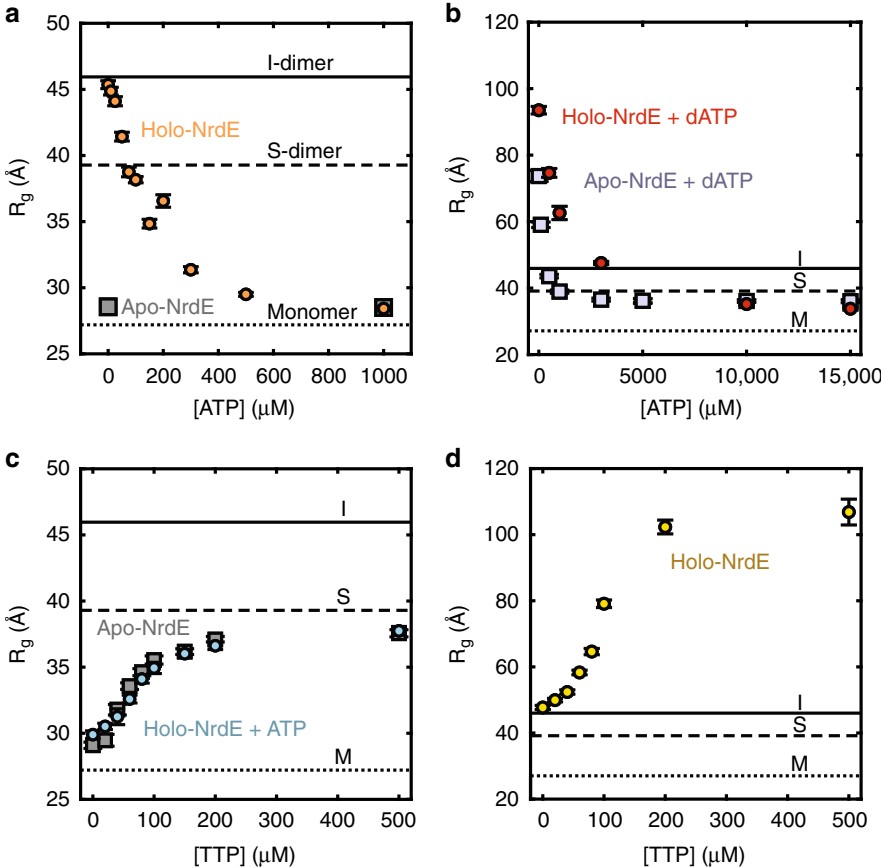

**Fig. 3** SAXS reveals a complex interplay of four distinct oligomerization states of NrdE. In all plots, the theoretical $R_g$ values of the I-dimer (46 Å, solid line), S-dimer (39 Å, dashed line), and monomer (27 Å, dotted line) are shown for comparison. **a** Titration of 0–1 mM ATP to 4 μM holo-NrdE (orange circles) leads to a reduction in $R_g$, consistent with the dissociation of I-dimer to monomer and confirmed by SEC–SAXS (Supplementary Fig. 7a). In contrast, the $R_g$ of 4 μM apo-NrdE (gray squares) remains constant up to 1 mM ATP. **b** Titration of 0–15 mM ATP to 4 μM holo-NrdE (red circles) or apo-NrdE (lavender squares) in the presence of 50 μM dATP leads to a decrease in $R_g$ that converges to a value near the theoretical value of an S-dimer. **c** Titration of 0–500 μM TTP to 4 μM apo-NrdE (gray squares) leads to an increase in $R_g$ that is suggestive of a monomer to S-dimer transition. A similar transition is observed with the addition of TTP to 4 μM holo-NrdE in the presence of 3 mM ATP (blue circles) and was further confirmed by SEC–SAXS (Supplementary Fig. 7b). **d** Titration of 0–500 μM TTP to 4 μM holo-NrdE in the absence of ATP leads to an increase in $R_g$ and a final profile that resembles the dATP-inhibited filament (Supplementary Fig. 2b, c). Source data are provided as a Source Data file

site of radical generation in NrdF (Y105) and the first RT residue in NrdE (Y684) (Fig. 2e, right), making it longer than the ~33 Å distance estimated for a class I RNR undergoing RT[34] (Supplementary Fig. 1c). However, in the absence of well-resolved density for the NrdF core, we cannot simply attribute the mechanism of inhibition to NrdF adopting a single conformation in which RT is prevented. Rather, we place significance on the way in which NrdF is confined within the helical interior (i.e. along the filament axis) of the NrdE filament. Density for the NrdF core suggests that they are closely packed relatively to each other (Fig. 2e, Supplementary Fig. 6b). Such confinement can still result in conformational disorder of the individual NrdF cores while hindering access to the full range of motions needed for NrdF to act as a catalytic partner for NrdE. In particular, limited motion of NrdF can inhibit RNR activity if sequential steps of enzymatic turnover require the sampling of multiple binding modes between subunits.

**Activation by ATP entails the dissociation of the I-dimer**. To investigate the structural basis for activation by ATP, we used SAXS to examine the effect of ATP on the dAMP-induced I-dimer. Titration of ATP into 4 μM holo-NrdE led to a reduction in $R_g$, indicative of I-dimer dissociation (Fig. 3a, orange circles).

Likewise, SEC–SAXS of holo-NrdE with 1 mM ATP and 0.5 mM CDP yielded a single scattering component that is well described by the theoretical scattering of a monomer[31] (Supplementary Fig. 7a). As a control, we confirmed that 4 μM apo-NrdE remains predominantly monomeric over the same range of ATP concentrations (Fig. 3a, gray squares). We thus find that at physiologically relevant concentrations, ATP disrupts the I-dimer interface.

We next examined the effect of ATP on the dATP-induced filament. Titration of ATP into 4 μM holo-NrdE pre-incubated with 50 μM dATP and 1 mM CDP led to a dramatic reduction in $R_g$ (Fig. 3b, red circles), indicating that the extended filament dissociates in the presence of ATP. However, rather than dissociating into monomers, $R_g$ converges to an intermediate value of ~36 Å, similar to the theoretical value for the S-dimer (39 Å). Apo-NrdE under comparably inhibited conditions converges to a similar $R_g$ upon ATP addition (Fig. 3b, gray squares). These results suggest that although ATP and dATP competitively bind the I-site, dATP remains bound at the S-site.

To differentiate nucleotide binding at the S-site and I-site, we performed titrations with thymidine 5′-triphosphate (TTP), a specificity effector that is expected to exclusively bind the S-site[30]. Addition of TTP into 4 μM holo-NrdE in the presence of 1 mM ATP (Fig. 3c, cyan circles) or into 4 μM apo-NrdE without ATP

 5

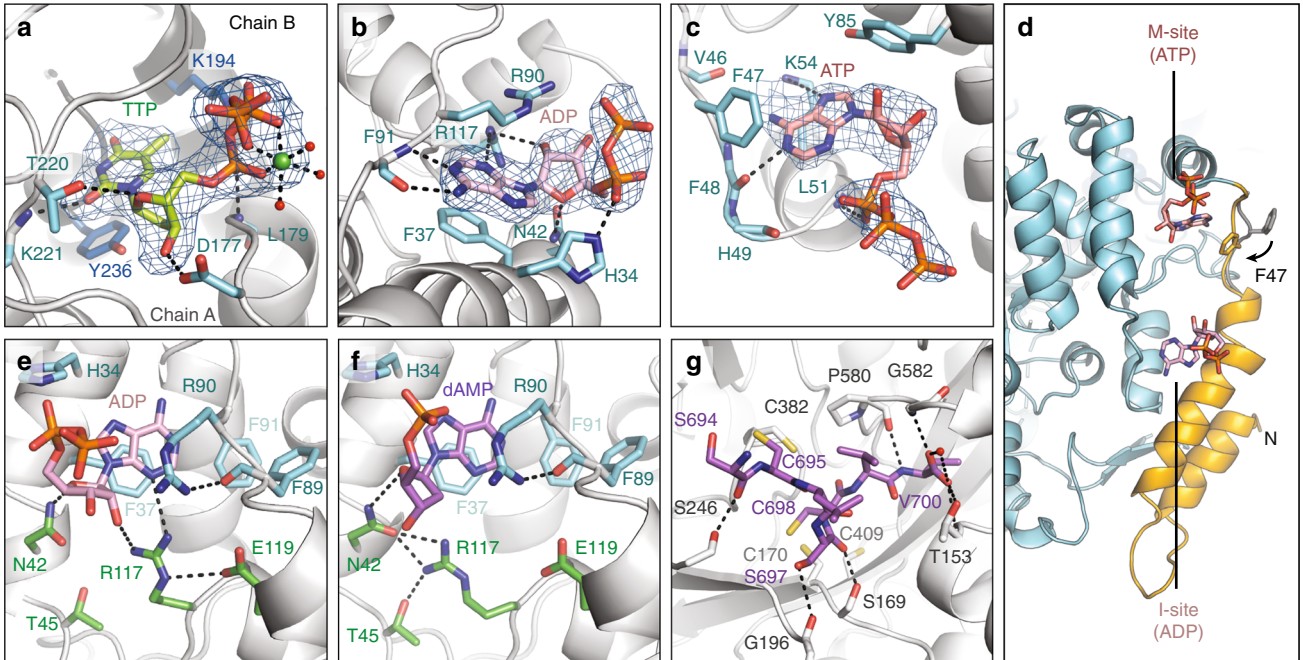

**Fig. 4** Crystallographic insight into allosteric activation and re-reduction of the catalytic site. **a–c** Shown in blue mesh are the $mF_O – DF_C$ Polder omit maps for the ligands in the 2.50 Å dataset, contoured at 4.0σ. **a** TTP binds to the S-site and interacts with both chains of the S-dimer interface. **b** ADP binds to the I-site, displacing dAMP that was previously observed to bind this site in holo-NrdE[31]. **c** ATP is bound to a newly identified M-site. **d** The M-site and I-site are both found in the N-terminus (orange) in close proximity to each other. Binding of ATP at the M-site induces F47 to flip inward (orange) relative to its position in the I-dimer (gray, PDB: 6CGL)[31]. **e** In our 2.50 Å structure, we observe R117 making H-bonds to the 2′-OH of the ADP ribose and E119. **f** With dAMP bound to the I-site, R117 instead H-bonds with N42 and T45 (PDB: 6CGL)[31]. In this position, R117 forms part of the I-dimer interface (Supplementary Fig. 9a). R117 is thus important for discrimination of ribonucleotides and deoxyribonucleotides at this site. **g** The NrdE C-terminus (purple sticks) was captured in the catalytic site of the 2.55 Å reduced dataset, revealing specific interactions between highly conserved residues. C698 on the C-terminal tail is captured in a conformation well poised to initiate re-reduction of a C170–C409 disulfide that forms in the catalytic site after each turnover (Supplementary Fig. 10c, d)

(Fig. 3c, gray squares) led to increases in $R_g$ suggestive of a monomer to S-dimer transition. Moreover, a three-state fit to an SEC–SAXS dataset collected on holo-NrdE with 1 mM ATP and 250 μM TTP showed that the entire elution can be decomposed as an interconversion of monomer and S-dimer with no contribution of the I-dimer (Supplementary Fig. 7b). In contrast, addition of TTP to 4 μM holo-NrdE in the absence of ATP induces the formation of a large species, akin to the dATP-induced filaments (Fig. 3d, Supplementary Fig. 2c). Combined, these results provide compelling evidence that specificity effectors, including dATP, favor the formation of the S-dimer interface and that ATP activates the enzyme by reversing the I-dimer-forming effects of dATP and dAMP.

**Crystal structures reveal molecular basis for ATP recognition.** With insight from SAXS (see the section "Methods"), a 2.50 Å resolution crystal structure of the S-dimer was obtained by co-crystallizing holo-NrdE with 5 mM $Mg^{2+}$-ATP, 0.5 mM TTP, and 1 mM GDP. Electron density at the S-site could be unambiguously modeled as TTP coordinating a $Mg^{2+}$ ion (Fig. 4a). Residues from both chains contribute to the S-site (Supplementary Fig. 8a), stabilizing the S-dimer.

Electron density at the I-site showed strong evidence for an ADP ligand displacing dAMP (Fig. 4b). As ADP was not present in the crystallization condition, we suspect that it is a hydrolysis product of ATP (see the section "Methods"). The bound ADP retains an H-bond to the α-phosphate and all of the adenine-specific interactions that were previously observed in dAMP-bound structures[31] (Supplementary Fig. 8b, d). Although no

interactions are observed with the β-phosphate, significant new interactions are observed involving the ribose. Specifically, R117 is within H-bonding distance of the 2′-OH of the ribose, N3 of the adenine ring, and E119 (Fig. 4e, green residues; Supplementary Fig. 8b). In the I-dimer structure (PDB: 6CGL), R117 points away from the dAMP deoxyribose and is instead anchored by H-bonds to N42 and T45 (Fig. 4f, green residues). This conformation appears to be important for the I-dimer interface, as it positions N42 and R117 to form H-bonding and cation–π interactions with F47 from the opposing chain (Supplementary Fig. 9a). Formation of the I-dimer interface additionally places the 3′-OH of dAMP within H-bonding distance of E53 of the opposing chain[31].

These observations suggest that the I-site is an adenine-specific site with the flexible side-chain of R117 acting as a sensor for the sugar identity. Interactions specific to deoxyribonucleotides appear to be made only when the I-dimer interface is formed, while those specific to ribonucleotides disfavor interactions with the F47-loop of a second chain. Such a mechanism also explains our SAXS results, which indicate that NrdE is a monomer in the absence of nucleotides at the I-site (Fig. 2a, b). In the crystal structure of apo-NrdE (PDB: 6CGM)[31], R117 mimics the ribonucleotide-bound conformation by retaining its interactions with E119 (Supplementary Fig. 9b).

Interestingly, we also observed nearby electron density for an ATP ligand with the adenine ring positioned against the F47-loop (Fig. 4c, d). However, compared with the analogous interactions seen at the I-site, the F47-loop appears better suited to interact with a guanine base (Supplementary Fig. 8c–f)[35]. We thus propose that GDP can bind this site and may in fact be the

preferred ligand. In support of this, we screened 27 diffraction-quality crystals grown with various combinations of ATP, TTP, GDP, and CDP, and only those grown with GDP displayed electron density at the M-site. For example, no density was observed at this site in a 2.95 Å structure of the S-dimer obtained with CDP instead of GDP in the crystallization condition (Supplementary Fig. 9c, d). The non-hydrolyzed nature of the ATP bound to this site further suggests that it was introduced when the crystal was soaked in freshly made cryoprotectant solution containing 5 mM ATP and that it may have displaced a GDP that primed this site for nucleotide-binding. Regardless, an important consequence of either ATP or GDP bound to the F47-loop is that the F47 side-chain flips inward by ~90° to form a π-stacking interaction with the purine ring (Fig. 4d). With F47 in this conformation, it is unable to participate in the formation of an I-dimer interface. We thus define this allosteric site as the "M-site" for its proposed ability to cause dissociation of I-dimers into monomers (Figs. 1a, 4d).

Together, our crystal structures and SAXS results suggest that inhibition by I-dimer formation requires two conditions to be met: the I-site must be loaded with a deoxyadenosine nucleotide, and the M-site must be empty in both chains. Consistent with this, our cryo-EM maps of the dATP-induced filament display empty pockets at the M-site (Supplementary Fig. 6c, purple stars). However, the absence of nucleotide in the M-site of our 2.95 Å S-dimer structure suggests that ATP's activating effect is primarily exerted by displacement of deoxyribonucleotides at the I-site.

**Visualization of the α C-terminus in the active site**. In all class I RNRs, the α C-terminus is thought to enter the catalytic site after each turnover to allow for the tail cysteines to regenerate the reducing cysteines via thiol-disulfide exchange[17] (Supplementary Fig. 1d). Unexpectedly, we were able to trap the final six residues of the NrdE C-terminus in the catalytic site of our 2.50 Å structure due to partial disulfide formation between the tail cysteines (C695, C698) and two catalytic-site cysteines (C382, C170) (Supplementary Figs. 10a, 11). These linkages, particularly that involving C695 and the initial site of the catalytically essential thiyl radical, C382, are likely artifacts of sample oxidation. However, they allowed us to capture the flexible C-terminus, which has never before been visualized in any class I RNR. Furthermore, we were able to use X-ray reduction of the same crystal to determine a 2.55 Å resolution structure depicting the reduced catalytic site (Fig. 4g, Supplementary Table 4, Supplementary Fig. 11). With the exception of the terminal residue V700, for which the side-chain does not participate in any polar interactions, all of the residues involved in binding the disulfide-trapped C-terminus are highly conserved in NrdE sequences (see the section "Methods") and nearly superimposable with corresponding residues in previous structures of NrdE with no C-terminus bound[29,31].

Re-reduction of the catalytic-site cysteines is expected to require two sequential $S_N2$ nucleophilic substitutions. Alignment of a *Salmonella typhimurium* NrdE structure[29] (Supplementary Fig. 10c, dark gray) with the C-terminus from our X-ray-reduced structure (Supplementary Fig. 10c, purple sticks) places C698 within 3 Å of the oxidized C178–C415 pair in the *S. typhimurium* catalytic site (equivalent to C170–C409 in *B. subtilis* numbering). These three cysteines form a triad with optimal geometry for disulfide exchange, with C698 well positioned to attack C170. Thus, the tail-binding interactions involving C698, C170, and C409 appear to be physiologically relevant for the first step of re-reduction, in turn likely resulting in the oxidized C698–C170 we observe in our disulfide-trapped structure. In contrast, C695 is too far from C698 to perform the second step of re-reduction

(Supplementary Fig. 10d) and is instead positioned closer to C382. A possible explanation is apparent in our X-ray-reduced structure (Fig. 4g), in which we resolve an additional interaction: an H-bond between the residue adjacent to C695 (S694) and a non-conserved residue near the catalytic site (S246). Under physiological conditions, we expect C695 to move closer to C698, pulling the remainder of the C-terminus further into the catalytic site.

Our structures also provide insight into the relationship between re-reduction and RT. In the current model for thiyl radical generation in class I RNRs, two tyrosines stack over C382 to form a π–π dyad required for co-linear proton-coupled electron transfer (PCET) (Supplementary Fig. 1c)[18,36–38]. These tyrosines (Y683/Y684) are located in the $β_J$-strand preceding the C-terminal tail sequence of NrdE. In structures of the class Ib RNR from *S. typhimurium*[29,33], this strand is observed forming a β-sheet with the adjacent $β_I$-strand (Supplementary Fig. 12a, yellow), whereas in ours, it is partially unzipped from the β-sheet, allowing the C-terminus to reach the catalytic site (Supplementary Fig. 12b, yellow). The Y684 side-chain flips out by ~180° to accommodate the resulting bend. These observations suggest that binding of the NrdE C-terminus is associated with the Y–Y dyad becoming unstacked, and hence, re-reduction of the catalytic site and RT cannot occur concurrently. Such a conformational gating mechanism would allow RNR to coordinate the multiple processes required for activity.

**Active NrdEF is a flexible tetramer**. To gain insight into the conformational ensemble of the active complex, NrdEF was examined with SEC–SAXS under various non-inhibiting conditions using a C382S mutant of NrdE to prevent changes in nucleotide concentrations (see the section "Methods"). We initially found that without both ATP and a specificity effector present, stoichiometric combinations of holo-NrdE and NrdF resulted in complex mixtures of oligomers that often include species that are larger than an $α_2β_2$ tetramer. Under S-dimer promoting conditions (1 mM ATP, 250 μM TTP), however, the scattering was dominated by a single species with a molecular weight estimate of 246 kDa[39], consistent with $α_2β_2$ (Fig. 5a, blue curve; Supplementary Fig. 13).

To interpret the scattering, we first considered the widely accepted symmetric $α_2β_2$ docking model[18] (Supplementary Fig. 1a) as a starting model. Using the program AllosMod-FoXS[40], missing residues were modeled, and simulated conformers consistent with the starting model were sampled to minimize the fit to the experimental scattering. A poor fit was obtained (Fig. 5a, purple dashed; $χ^2 = 20.43$), indicating that the solution ensemble cannot be captured by the local energy landscape of the docking model (Fig. 5b, bottom). A similarly poor result was obtained from an "expanded" $α_2β_2$ starting model with a pronounced gap at the subunit interface generated by fitting a NrdF dimer crystal structure (PDB: 4DR0)[21] into the $α_2β_2$ ASU of the NrdEF cryo-EM map (Fig. 2e, right and Fig. 5a, orange dashed; $χ^2 = 18.55$). We thus considered an alternative $α_2β_2$ conformation depicted by a 4.0 Å resolution crystal structure of the class Ib RNR from *S. typhimurium*[33] in which $β_2$ is angled asymmetrically with respect to the $α_2$ S-dimer, exposing the catalytic sites (Supplementary Fig. 1b). Remarkably, this starting model resulted in the best fit (Fig. 5a, black dashed; $χ^2 = 2.25$). The final model remains asymmetric overall, but the angle between the two subunits is reduced by 14° (Fig. 5b, top). Thus, although *B. subtilis* RNR must sample a compact conformation for long-range RT, our analysis suggests that an asymmetric arrangement is a significant component of the conformational ensemble in solution.

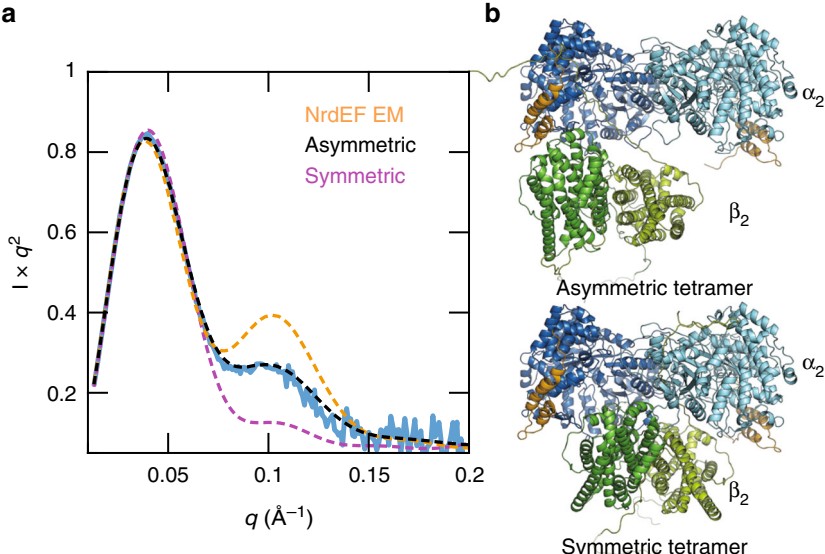

**Fig. 5** The active NrdEF structure is a flexible $\alpha_2\beta_2$. **a** SEC–SAXS of a stoichiometric mixture of C382S holo-NrdE and NrdF under activating conditions (1 mM ATP, 250 μM TTP) revealed a predominant species (experimental profile shown in blue) with a molecular weight estimate[39] of 246 kDa, consistent with an $\alpha_2\beta_2$ species (actual MW of 243 kDa). Structural modeling was performed in AllosMod-FoXS[40] using three different $\alpha_2\beta_2$ starting models. Using a symmetric $\alpha_2\beta_2$ docking model[18] as a starting model underestimated the shoulder observed in the mid-$q$ region (purple dash; $\chi^2 = 20.43$), suggesting that the solution structure is more open. In contrast, the expanded $\alpha_2\beta_2$ starting model derived from the ASU of the NrdEF cryo-EM structure (Fig. 2e, right) overestimated the shoulder (orange dash; $\chi^2 = 18.55$), suggesting that the subunits are closer together in solution under activating conditions. An asymmetric $\alpha_2\beta_2$ starting model based on an *S. typhimurium* structure (PDB: 2BQ1)[33] yielded the best-fit conformer (black dash; $\chi^2 = 2.25$). SAXS profiles are shown in Kratky representation ($q$ vs. $Ixq^2$) to emphasize mid-$q$ features. **b** Best-fit models of the asymmetric $\alpha_2\beta_2$ (top) and symmetric $\alpha_2\beta_2$ (bottom). Source data are provided as a Source Data file

## Discussion

Based on our SAXS, EM, and crystallography results, we propose an allosteric model for *B. subtilis* RNR involving six distinct NrdE-containing species (Fig. 6). At physiologically relevant protein concentrations (low μM), nucleotide-free NrdE is primarily monomeric. The I-site is specific to adenine nucleotides where binding of deoxyribonucleotides (dAMP and dATP in particular) leads to a partially inactive I-dimer. In contrast, binding of any specificity effector (dATP, dGTP, or TTP) to the S-site stabilizes the S-dimer interface. Thus, the combination of these nucleotides leads to a NrdE filament having alternating dimer interfaces, with the inhibitor dATP uniquely able to bind both sites. Although NrdF is able to associate with the NrdE filament through its C-termini, when confined within the helical interior, we propose that it is unable to access the sequential motions necessary for interaction with NrdE as a catalytic partner. Conversely, the primary activating effect of ATP is to destabilize the I-dimer interface by displacing deoxyribonucleotides at the I-site. As ATP has a low affinity for the S-site, the combination of a specificity effector at the S-site and ATP at the I-site favors S-dimer formation. Likewise, ATP is able to dissociate the NrdEF filament at the I-dimer interfaces, producing $\alpha_2\beta_2$ complexes as the active form. In this oligomerization state, NrdF is able to access the full range of motions needed for RNR activity.

Together with prior work[31], our results indicate that two unique allosteric sites have evolved in *B. subtilis* RNR. In particular, the I-site, which is formed by residues in the $\alpha_2$-helix of the truncated ATP-cone and adjacent secondary structural elements, has evolved adenine specificity and sugar discrimination (Fig. 4, Supplementary Fig. 8). However, it shows no apparent discrimination for the number of phosphates, perhaps indicating that unlike the S-site, it has yet to evolve specificity for triphosphates, the final products of deoxyribonucleotide metabolism. Alternatively, the ability of the I-site (but not S-site) to tightly bind dAMP in *B. subtilis* RNR[31] may be advantageous as an

additional tuning dial for activity regulation. Although not shown in our scheme (Fig. 6), our results also suggest a secondary mechanism for activation involving the M-site. Because the M-site is coupled to the F47-loop, ATP and possibly GDP binding here would directly interfere with I-dimer formation. Notably, it has been shown by analytical ultracentrifugation that in the presence of GDP, *B. subtilis* NrdE is monomeric[12]. However, as apo-NrdE is preferentially a monomer, binding of nucleotide at the M-site does not appear to be a requirement of I-dimer dissociation.

Our crystal structures provide key insight into the requirements of an active complex. The $\alpha$ C-terminal residues that we observe (695-CLSCVV-700) occupy the catalytic site in a conformation that is remarkably similar to the binding mode of nucleotide substrates observed in other class I RNRs, with the terminal carboxylate mirroring the interactions of a substrate $\beta$-phosphate (Supplementary Fig. 10a, b)[19]. Since catalytic-site reduction by the $\alpha$ C-terminus is likely a derived trait of class I and II RNRs[41], this similarity suggests that the C-terminus evolved to mimic substrate binding. These structures also imply that when the $\alpha$ C-terminus is bound, RT cannot occur due to the coupled unstacking of the Y–Y dyad. Conversely, alignment of our structures with the *S. typhimurium* $\alpha_2\beta_2$ structure[33] suggests that when the Y–Y dyad is stacked, the associated $\beta$-sheet (Supplementary Fig. 12b, yellow cartoon) presents a grooved surface that may serve as a potential binding site for the $\beta$ C-terminus. Such a binding mode would bring Y307 on the flexible region of the $\beta$ C-terminus in close proximity to the Y–Y dyad in an arrangement thought to be necessary for RT (Supplementary Fig. 1c). Together, these structures imply that binding of the $\alpha$ and $\beta$ C-termini to the catalytic site may be mutually exclusive and that significant structural dynamics are required for RNR activity.

Consistent with this notion, our SAXS results provide evidence for an $\alpha_2\beta_2$ conformation that is asymmetric like the *S.*

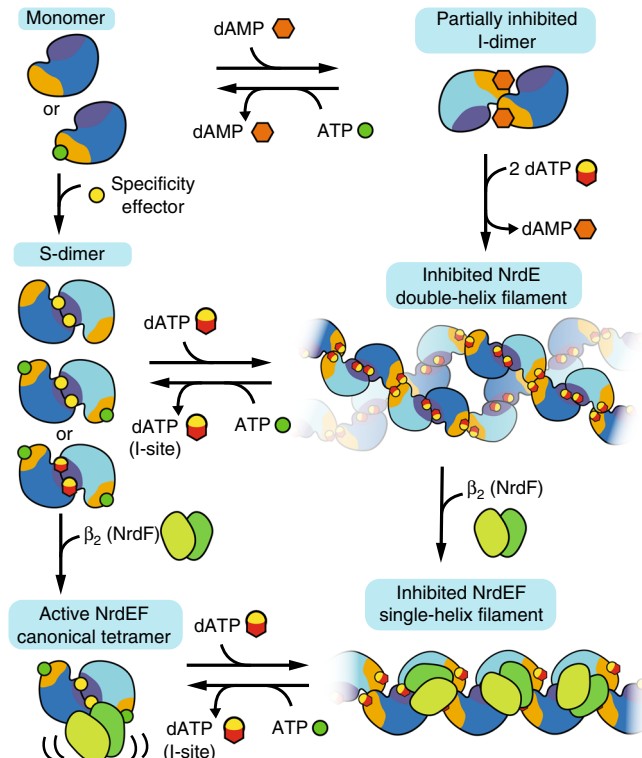

**Fig. 6** A model for the overall allosteric regulation of *B. subtilis* RNR. Without nucleotides, NrdE is a monomer, but dAMP and dATP can bind the I-site and induce a partially inhibited I-dimer. Addition of specificity effectors instead induces the monomer to form an S-dimer. When specificity effectors (including dATP) bind to the I-dimer, they induce formation of an inhibited double-helical NrdE filament composed of alternating S-dimer and I-dimer interfaces. NrdF competes for the NrdE double-helical interface, and thus NrdF binding leads to the dissociation of the NrdE double-helix into individual helical structures. NrdF binds to the helical interior of the NrdE filament, leading to an inter-subunit gap and positional confinement that prevent NrdF from accessing NrdE for turnover. Both the NrdE and NrdEF filaments are reversible by addition of ATP, which can displace dATP from the I-site and induce dissociation of the I-dimer interface. Finally, addition of NrdF to the S-dimer leads to formation of an active but asymmetric $\alpha_2\beta_2$ tetramer in which a hinge motion between the two subunits plays an important role in activity

*typhimurium* NrdEF structure[33]. In this structure, one lobe of NrdF makes direct contact (570 Å$^2$ buried surface area) with two notable structural features of NrdE: the first helix of the truncated ATP-cone, as well as a β-hairpin loop with potential evolutionary significance (Supplementary Fig. 1b, inset). The involvement of the truncated ATP-cone suggests that it may have a role beyond I-dimer formation that is important for activity. Additionally, the β-hairpin loop, which has a known function in class II RNRs for binding the adenosylcobalamin radical cofactor[42], perhaps serves a similar role of stabilizing the radical subunit in class I RNRs. The resulting asymmetric interactions of $\beta_2$ to $\alpha_2$ may enable one side of the tetramer to engage in RT while the other catalytic site remains open for re-reduction.

Although filaments of other enzymes in nucleotide metabolism have recently been reported[25,43,44], the dATP-inhibited filament of *B. subtilis* RNR is unusual for its ability to modulate catalytic activity by reversible assembly. To date, no other RNR lacking an ATP-cone has been reported to display such behavior. Consistent with this, the residues involved in the I-site and F47-loop are fully conserved in only a small subset of class Ib RNRs[12,31,45] (see the section "Methods"). We thus hypothesize that truncation of the

ATP-cone in the class Ia progenitor of class Ib RNRs rendered the domain vestigial and that the original class Ib RNR did not need activity regulation because it was the organism's secondary RNR and was only expressed under iron-poor conditions[46]. Because the class Ib enzyme of *B. subtilis* is the sole RNR used by the organism, evolution of new binding sites and allosteric mechanisms within the truncated ATP-cone domain may have been advantageous. However, as class Ib RNRs are used by a number of pathogens, it is also worth noting that loss of RNR allostery, which has been linked to increased mutation rates[47–49], may also provide a selective advantage in evolving therapeutic resistance[50].

Perhaps the most striking result of this study is that with *B. subtilis* RNR, nature has arrived at a remarkably similar solution to the ATP-cone, whereby binding of dATP induces a new oligomeric interface that disrupts the protein–protein interactions needed for RT between α and β. The convergence of these two evolutionary routes with disparate origins reveals the importance of nucleotide homeostasis as a selective pressure, as well as the flexibility by which new protein–protein interactions can evolve.

## Methods

**Expression and purification of *B. subtilis* RNR proteins**. Protein expression and purification were performed following established protocols[30,31]. Tagless NrdE (both wild type and C382S variants) was produced using SUMO fusion technology with pE-SUMO-*nrdE* vectors developed in a previous study[12,31]. His$_6$-Smt3-tagged NrdE was overproduced in BL21 (DE3) cells and purified with a nickel or cobalt affinity column followed by anion exchange chromatography on a Q-sepharose column. The His$_6$-Smt3 (SUMO) tag was cleaved using SUMO protease and separated from the tagless NrdE protein with a second nickel affinity chromatography step. To separate nucleotide-free (apo-NrdE) and dAMP-bound (holo-NrdE) fractions, a final high-resolution anion exchange chromatography step was conducted using a MonoQ 10/100 GL column (8 mL). His$_6$-tagged apo-NrdF was produced in a similar manner but without the tag cleavage steps and with the addition of 100 μM 1,10-phenanthroline in the cell lysis buffer to prevent mis-metallation. Apo-NrdF was then reconstituted with Fe(III)$_2$–Y• or Mn(III)$_2$–Y• according to previously established protocols[12,30,31]. Briefly, iron reconstitution was performed by anaerobic addition of $(NH_4)_2Fe(SO_4)_2$ to apo-NrdF followed by incubation with oxygen-saturated buffer. For manganese reconstitution, $MnCl_2$ is instead used, concomitant with the hydroquinone form of the accessory protein NrdI, to generate the active Mn cofactor. In both cases, holo-NrdF is purified from apo- and mismetallated NrdF by a subsequent anion exchange chromatography step using a MonoQ column[12,30,31]. All protein concentrations are given as monomer concentrations. Unless otherwise noted, all studies were done in the standard assay buffer: 50 mM HEPES pH 7.6, 150 mM NaCl, 15 mM MgCl$_2$, 1 mM tris(2-carboxyethyl)phosphine (TCEP), and 1% or 5% (w/v) glycerol.

**Sequence conservation**. A multiple sequence alignment of NrdE sequences from the RNRdb[45] was performed in Muscle[51] and analyzed in matlab (The Mathworks). Of the 5216 non-redundant sequences, 151 (3%) show full conservation of residues involved in the I- and M-sites (H34, F37, N42, T45, F47, F48, H49, L51, 87-KKFRFP-92, R117, E119). In contrast, 4477 (86%) sequences show full conservation of residues identified in α C-terminus binding (T153, S169, G196, P580, G582, S697).

**Small-angle X-ray scattering**. X-ray scattering experiments were performed on 11 different occasions at the Cornell High Energy Synchrotron Source (CHESS) G1 station using a 250 μm square X-ray beam with an energy of 9.8–9.9 keV and flux of ~10$^{12}$ photons s$^{-1}$ mm$^{-1}$ at the sample position (Supplementary Tables 1 and 2). Small-angle and wide-angle X-ray scattering (SAXS/WAXS) images were simultaneously collected on two Pilatus 100K detectors covering a range of $q \approx$ 0.01–0.7 Å$^{-1}$. Here, the momentum transfer variable is defined as $q = 4\pi/\lambda \sin\theta$, where $\lambda$ is the X-ray wavelength and $2\theta$ is the scattering angle. Data processing at the beamline was performed in BioXTAS RAW[52]. Final data processing and analysis were performed using ATSAS[53] and in matlab following established protocols[32,54]. Briefly, scattering images were integrated about the beam center and normalized by transmitted intensities measured on a photodiode beamstop. The integrated protein scattering profile, $I(q)$, was produced by subtraction of background buffer scattering from the protein solution scattering. Radii of gyration ($R_g$) were estimated with Guinier analysis, and pair distance distribution analysis was performed in GNOM[55]. Error bars associated with $R_g$ values are curve-fitting uncertainties from Guinier analysis. Molecular weight estimation was performed using a Porod invariant method implemented in SAXSMoW[39,56].

For subunit and nucleotide titration experiments, proteins were first studied at multiple concentrations to examine inter-particle effects and oligomerization. Apo-NrdE (both wild-type and C382S) remained monomeric over a wide range of

concentrations (1–20 μM). Holo-NrdE displayed concentration-dependent oligomerization, with C382S having a greater tendency to dissociate than wild-type, likely due to having a lower dAMP content following purification. Under saturating nucleotide conditions, both constructs behaved identically. A near-physiological protein concentration of 4 μM[12,28] was chosen for subsequent titration to minimize concentration effects while maintaining a reasonable signal-to-noise ratio. Where appropriate, 1 mM CDP was included as a corresponding substrate (Supplementary Table 1)[25,28]. For all SAXS experiments where NrdE and NrdF were combined, a C382S mutant of holo-NrdE was used. Due to the quantity required, the subunit titration was performed with Fe-NrdF, which is structurally interchangeable with Mn-NrdF and can be produced with greater yield. For all titration experiments, background subtraction was performed with carefully matched buffer solutions containing identical concentrations of nucleotides following established protocols[54]. For each measurement, 40 μL of sample were prepared fresh and centrifuged at 14,000×g at 4 °C for 10 min immediately before loading into an in-vacuum flow cell kept at 4 °C. For each protein and buffer solution, 20 × 2 s exposures were taken with sample oscillation to limit radiation damage then averaged together to improve signal. SVD was performed in MATLAB.

Size-exclusion chromatography-coupled SAXS (SEC–SAXS) experiments were performed using a GE Superdex 200 Increase 3.2/300 (2.4 mL), Superdex 200 5/150 GL (3 mL), or Superdex 200 Increase 10/300 GL (24 mL) column operated by a GE Akta Purifier at 4 °C with the elution flowing directly into an in-vacuum X-ray sample cell (Supplementary Tables 1 and 2). To account for a ~10-fold dilution of the sample during elution, 50–75 μL samples were prepared with 40–80 μM protein in assay buffer with nucleotides. Samples were then centrifuged at 14,000×g for 10 min at 4 °C before loading onto a column pre-equilibrated in a matched buffer. Samples were eluted at flow rates of 0.05–0.1 mL min⁻¹ for the 3.2/300 column and 0.15–0.5 mL min⁻¹ on the 5/150 and 10/300 columns. For each sample, 2-s exposures were collected throughout elution until the elution profile had returned to buffer baseline, and scattering profiles of the elution buffer were averaged to produce a background-subtracted SEC–SAXS dataset. Data were analyzed by SVD and EFA using custom MATLAB code[32]. EFA allows for mathematical separation of partially co-eluting species, such as exchanging oligomeric species. Briefly, this process involves iterative SVD to identify changes in the number of distinguishable eluting species followed by decomposition of the dataset by alternating least-squares.

Structural modeling was performed using the ATSAS package[53] and AllosModFoXs[40,57] following previously established protocols[25,28,32]. Theoretical scattering curves were calculated in CRYSOL[58] with 50 spherical harmonics, 256 points between 0 and 0.5 Å⁻¹, and the default electron density of water. The overall scale factor and solvation parameters were determined by fitting to EFA-separated scattering curves. Multi-state fitting was performed in OLIGOMER[53,59] using form factors generated by FFMaker with 50 spherical harmonics and 201 points between 0 and 0.5 Å⁻¹. Form factor calculations in CRYSOL and FFMaker were performed with the following models: apo-NrdE monomer (PDB: 6CGM)[31], I-dimer (PDB: 6CGL)[31], S-dimer (PDB: 6MT9, obtained in this study), and a 34-mer model of the NrdE filament (PDB: 6MYX, obtained in this study). Geometric analysis of the NrdE filament was performed by comparing the experimental scattering to the known form factor for a hollow cylinder[60]. The α₂β₂ starting models for AllosModFoXs[40,57] were constructed by aligning a structure of the NrdE S-dimer determined in this study (PDB: 6MT9) and the NrdF dimer structure (PDB: 4DR0)[21] to a symmetric docking model of E. coli class Ia RNR[28] and an asymmetric structure of NrdEF from S. typhimurium class Ib RNR (PDB: 2BQ1)[33] (Supplementary Fig. 1a, b), as well as by rigid-body modeling into the asymmetric unit (ASU) of the NrdEF cryo-EM map (Fig. 2e, Supplementary Fig. 6b). Disordered and missing residues were modeled and sampling of static structures consistent with the starting model was performed in AllosModFoXs[40,57].

**Electron microscopy of the NrdE filament.** Samples of the dATP-induced NrdE filament were produced by incubating 40 μM holo-NrdE with 100 μM dATP and 1 mM CDP in assay buffer. The mixture was then diluted to 10 μM NrdE in the same nucleotide-containing buffer without glycerol, leaving a final glycerol concentration of <0.2% (w/v). Specimens were prepared by applying 5 μL of sample to a glow-discharged Quantifoil R1.2/1.3 holey-carbon cryo-EM grid, blotted for 3.5 s with Whatman #1 filter paper (GE Healthcare), and then plunge-frozen in liquid ethane using an FEI Vitrobot held at 90% humidity. The cryo-EM data were collected on a Talos Arctica 200 kV electron microscope (FEI). A representative image is shown in Supplementary Fig. 3a. In total, 1,158 movie images were recorded on a Gatan K2 Summit direct electron detector (in counting mode) at 1.515 Å per pixel, five frames per second, and 20 s exposures, resulting in 100 frames per movie and 22 e⁻Å⁻² total dose in each region of interest. Imaging defocus was varied in the range of 1.0–3.0 μm.

The density map of the NrdE helical filament was reconstructed using a single-particle method, and the data were processed in the PARTICLE package[61] (www.sbgrid.org/software/titles/particle). Micrograph movie-frames were first aligned via full-frame registration. Contrast transfer function (CTF) parameters of each micrograph were then determined by matching the theoretical CTF with the image Thon-rings in the 8–35 Å resolution range. All NrdE filaments were manually annotated using the "filament selection" tool in PARTICLE, which was used to mark both ends of a filament and then box out individual frames along the filament axis at 40 Å offset steps and at 256 × 256 pixels per frame. In total, 85,532 NrdE "particles" were boxed for 2D classification. The "direct particle classification" algorithm (DPC)[62] was then used in 2D particle classification.

PARTICLE organizes consecutive frames boxed from the same segment into one group to track the sequential connectivity among adjacent class averages and then pieces together a composite helical filament with enhanced image contrast and signal-to-noise ratio for initial 3D reconstruction (Supplementary Fig. 3b). Computationally, two neighboring class-average frames are composed together by aligning the overlapping regions of interest (ROIs) via local cross-correlation maximization using both translational and rotational searches. A small in-plane rotational degree-of-freedom was included in the NrdE filament refinement to counter potential local bending of the filament but was limited to <10° variation because the filaments are relatively rigid. Once aligned, the pixel intensity in the region of overlap is set to the average of the ROIs from both segments. This process is repeatedly applied to include all related class-averages to form the final composite filament. We found that the length of a repeating pattern measured in the composite filament is approximately 675 Å (Supplementary Fig. 3b, orange dashed line). Because the long dimension of a NrdE I-dimer is ~75 Å, according to the previously determined crystal structure (PDB: 6CGL), we predicted that this repeating pattern would contain nine dimeric units over a span of 675 Å. We thus estimated the unit rotation angle to be around $N \times 360°/9 = N \times 40°$, where $N = 1$, 2, 3… is the number of helical turns. A test of helical reconstruction based on these initial parameters quickly converged to ~75 Å unit rise and 80° unit rotation (where $N = 2$), which served as the initial helical parameter in the subsequent reconstruction and refinement.

The first step of refinement employed DPC class averages as the "particle stack" input. Taking the above initial model as reference, a projection-matching algorithm was used to align all DPC class averages and also used in the subsequent 3D reconstruction and refinement. According to the DPC algorithm[62], each "seed" raw image receives its initial alignment from the corresponding class average for the subsequent processing. A phase-residual-based projection-matching algorithm was used in the particle alignment. In each refinement cycle, the alignment parameters (X/Y-translation and three Euler angles) of each particle frame was adjusted to maximize the phase agreement with the current reference model projection up to the resolution cut-off at a Fourier shell correlation (FSC) of 0.143. In the 3D reconstruction, each particle frame was weighted by its phase-residual score in Fourier-space insertion. After 50 iterations, the 3D reconstruction converged to ~5.8 Å resolution (Supplementary Fig. 3c). To further improve the resolution of the helical reconstruction, the helical parameters were re-evaluated at the end of each iteration of refinement using the "helical symmetry solver" function in PARTICLE, in which a local search (within 2 Å and 2°) around the current helical parameters was performed to identify the maximum density cross-correlation within a cylindrical mask over two repeating unit (150 Å) in the middle of the double-helix density map (Supplementary Fig. 3d). The resulting helical parameters were applied to symmetrize the reconstructed density map in the current iteration, which then served as the reference for the next iteration of refinement. Through 50 iterations of masked local refinement, the 3D reconstruction converged to 4.8 Å resolution as determined by the standard 0.143 FSC criterion. However, based on features of our map as well as our map-model FSC, we believe the NrdE map to actually be closer to 6 Å and thus report it as such. The double-helix NrdE filament has two sets of helical symmetry parameters: the inter-helix symmetry (74.24 Å, −81.28°, where the negative sign indicates a flip in handedness) and intra-helix symmetry (37.12 Å, 139.36°).

**Electron microscopy of the NrdEF filament.** Cryo-EM samples of the NrdEF filament were prepared by mixing 20 μM holo-NrdE with 20 or 40 μM Mn-reconstituted NrdF in assay buffer with 100 μM dATP and 1 mM CDP, prior to four-fold dilution with nucleotide-containing buffer. 3.5 μL of the diluted sample (5 μM NrdE and 5 or 10 μM NrdF) was applied to glow-discharged 200 mesh C-flat grids (Protochips, 2 μm hole size), blotted with Whatman #1 filter paper (GE Healthcare), and plunge-frozen in liquid ethane using an EM-GP (Leica) held at 95% relative humidity. A subset of the grids was pre-coated with a support film of continuous, amorphous carbon by flotation of cleaved mica. For these grids, the sample was diluted to a final protein concentration of 2 μM. Data were collected on a 200 kV Talos Arctica cryo-electron microscope (FEI). Particles from 2843 images were used in the final reconstruction, of which 446 images were obtained over continuous carbon film. Images were recorded using a K2 Summit camera (Gatan) operating in counting mode: nominal 1.05 Å per pixel, five frames per second, with flux ranging from 5 to 12 e⁻ per pixel per second. 95% of images used had defocus between 0.8 and 2.5 μm. A representative image is shown in Supplementary Fig. 4a.

Movie frames were aligned with MotionCor2[63]. CTF parameters were fit with Gctf[64]. Filaments were manually annotated and extracted using the helix boxer in RELION[65,66]. The annotator selected regions, where filaments were straight and not contacting other filaments, aggregates, or contaminants. For extraction, the filaments were divided into boxes every 64 pixels (67.3 Å). Particles were imported into cryoSPARC v2[67] for further processing. An initial model was generated ab initio by cryoSPARC's stochastic gradient descent procedure that uses a random initialization. With this initial model as a reference, the full dataset was refined to 4.99 Å and then further refined to 4.65 Å by incorporating non-uniform 3D refinement. Finally, the reconstruction was masked and locally filtered based on the local resolution estimate. We did not observe significant differences in NrdE

internal structure when 2D classification was used to select a subset of particles for analysis, but the resolution was 0.07 Å worse. It is possible that data sorting was superfluous because manual boxing of helices pre-selects for high-quality particles. Because cryoSPARC v2 does not support helical symmetry, we used a large box size (480 pixels) containing many NrdEF repeats for processing as an asymmetric single particle. Helical symmetry emerged naturally in the $C_1$-refined map. We then re-cropped the final reconstruction to display the central NrdEF repeats to avoid radial and/or edge effects near the box boundaries. The handedness of the final map was flipped based on the handedness of published NrdE structures[31].

To test for consistency, the full dataset was also processed in cryoSPARC v1[67] and in RELION[66], with a box size of 256 pixels and considering helical symmetry in the case of RELION. At the resolution of the various output maps, we did not observe notable differences in NrdE internal structure from package to package. We also simulated several biased models using UCSF Chimera[68]. These include a NrdE-only map made by erasing NrdF density using SEGGER[69]; a filament based on an asymmetric *S. typhimurium* NrdEF crystal structure (PDB: 2BQ1)[29] (Supplementary Fig. 1b) and simulating electron density with *molmap*; and a filament based on a symmetric docking model[18] (Supplementary Fig. 1a), also simulated with *molmap*. 3D autorefinement was performed in RELION using these simulated maps as references to test whether a biased reference would cause refinement to converge to a different local minimum. In all cases, refinement converged on a map with a pronounced gap at the subunit interface. The maps resulting from biased models were not carried forward to any other refinement, and no initial models were used in creation of the deposited NrdEF map.

The final helical parameters were consistent from package to package. The helical twist measured at 88.6° (cryoSPARC v2) or 90.9° (RELION), and helical rise was measured at 73.8 Å (cryoSPARC v2) or 73.2 Å (RELION).

**Electron microscopy model refinement**. Model refinement was first performed with the higher-resolution map of the dATP-induced NrdEF filament. An initial model for the NrdE monomer was built using the crystal structure of the dAMP-bound I-dimer (PDB: 6CGL)[31] as the basis. Residues involved in forming the S-dimer interface (175–238, 289–388) were then modeled using the 2.50 Å structure of the S-dimer as the template. Four copies of the monomer were docked into the map in the UCSF Chimera package[68]. The NrdF C-terminus was modeled using the 4.0 Å *S. typhimurium* NrdEF (PDB: 2BQ1)[33] as a reference. Due to the low resolution of both the map and the template structure, we did not attempt to assign the residues and instead modeled the C-terminus as an eight-residue polyalanine chain. dATP molecules at the I- and S-sites were modeled using the dAMP-bound structure (PDB: 6CGN)[31] and the 2.50 Å structure (PDB: 6MT9)[31] as references. Manual adjustment in Coot[70] was performed iteratively in combination with real space refinement in Phenix[71] using global minimization, secondary structure restraints, non-crystallographic symmetry (NCS) constraints, and atomic displacement parameter (ADP) refinement. Geometry and model statistics were evaluated with MolProbity[72] and the wwPDB validation server. Density for the NrdF core displayed considerably weaker definition (Supplementary Fig. 5b, d), and the core was therefore not included in the deposited model. Additionally, EM density was observed near C382 that could correspond to CDP, although the density map was not sufficiently defined to allow reliable modeling. The final model of the ASU contains a NrdE S-dimer, with the NrdF C-terminal peptide bound in each monomer and a dATP molecule at each S- and I-site.

The dATP-induced double-helical NrdEF filament was modeled by fitting four copies of an α monomer from the NrdEF cryo-EM structure into an ASU consisting of two S-dimers from opposing interacting helical strands. The structure was refined via Phenix real-space refinement, but manual refinement of sidechains was not pursued due to the lower resolution of the map.

Data refinement and model statistics are provided in Supplementary Table 3. Estimates for σ levels corresponding to the absolute thresholds used to contour maps are shown in Supplementary Table 5. We note that the standard deviations from the mean value are statistically meaningful in the unmasked, unfiltered half-maps but not in the locally filtered final map after solvent has been masked out. To estimate the former, we determined thresholds in the half-maps that corresponded to that of the final map and computed the σ levels. For both EM maps, we report a global resolution determined where FSC = 0.143 between two independently refined half-maps[73] (Supplementary Fig. 5c, d). Local resolution maps calculated in ResMap[74] are shown in Supplementary Fig. 5a, b. Model resolution was determined where the map-model FSC = 0.5 (Supplementary Fig. 5e, f). The atomic coordinates and maps of both the NrdE and NrdEF filaments have been deposited to the Protein Data Bank and EM Data Bank under accession codes PDB 6MYX and 6MW3 and EMD-9293 and EMD-9272. Figures were made in UCSF Chimera[68], and difference maps shown in Fig. 2g, h were computed in Phenix[75] (phenix.real_space_diff_map) using a model of α_4 (composed of two S-dimers associating at the I-dimer interface) with ligands removed.

**Crystallography**. Crystallization trials were first conducted with holo-NrdE in the presence of 5 mM Mg$^{2+}$-ATP and 1 mM CDP. Optimization of a condition that yielded small crystal bundles led to the growth of larger crystals (~50 μm in size) in 8 days, one of which yielded a 5.6 Å diffraction data set in $P2_12_12_1$. Molecular replacement using a monomeric NrdE structure (PDB: 6CGM)[31] as the search model yielded two chains in the ASU that formed a canonical S-dimer. Since our

SAXS results had suggested that ATP is a poor specificity effector, we speculated that we could stabilize this packing arrangement by addition of TTP to the previously noted crystallization condition. Large crystals (~100 μm in size) having the same crystal form ($P2_12_12_1$) were obtained in ~4 days when 4.5 mg mL$^{-1}$ holo-NrdE was co-crystallized with 5 mM Mg$^{2+}$-ATP, 0.5 mM TTP, and 5 mM CDP. Use of the preferred substrate for TTP (1 mM GDP) in place of CDP led to the appearance of large crystals in 2 days. These crystals have the same packing arrangement except with higher crystal symmetry ($P4_32_12$) and one monomer in the ASU, which forms an S-dimer with a symmetry molecule. Thus, systematic changes to the nucleotide mixtures led to progressive improvement in crystalline order and, in turn, resolution. Crystals for data collection were grown by vapor diffusion from 4.5 mg mL$^{-1}$ tagless holo-NrdE in 50 mM HEPES pH 7.6, 50 mM NaCl, 5 mM MgCl$_2$, 2 mM TCEP, and 1% glycerol, supplemented with 5 mM ATP, 500 μM TTP, and either 1 mM GDP or 5 mM CDP. The protein solution was incubated for 10 min with freshly added nucleotides prior to being mixed in a 1:1 hanging drop with a precipitating solution of 6% PEG 3350, 1% w/v tryptone, and 50 mM HEPES at pH 6.9 or 7 for crystals grown in the presence of CDP and GDP, respectively. Crystals were cryoprotected by soaking for 5–10 s in well solution with 8% w/v sucrose, 2% w/v glucose, 8% v/v glycerol, 8% v/v ethylene glycol and supplemented with nucleotides, TCEP, and MgCl$_2$ adjusted to the same concentrations used in the original protein solutions.

Data collection was performed at CHESS beamline F1 on a Pilatus 6M detector at 100 K, a wavelength of 0.9775 Å, and with 0.2° oscillation step. Diffraction images were integrated using either XDS[76] ($P4_32_12$) or iMosflm[77] ($P2_12_12_1$) and were scaled and merged using AIMLESS[78]. Phases were estimated by molecular replacement in Phaser[79] using the apo-NrdE monomer structure (PDB: 6CGM)[31] as the search model. Model building in Coot was performed iteratively with positional and B-factor refinement in Phenix[75]. Model geometry was analyzed with the MolProbity server and the wwPDB validation server. Data collection and model refinement statistics are shown in Supplementary Table 4. Figures were generated using PyMol (Schrödinger, LLC). Polder omit maps were generated using Phenix[75]. Ligand interaction diagrams were generated using LigPlot[80]. The atomic coordinates and structure factors have been deposited in the Protein Data Bank under accession codes 6MT9, 6MVE, and 6MV9.

For the crystal structure of the NrdE S-dimer with an empty M-site (PDB: 6MV9), a dataset collected on a crystal grown from the TTP/ATP/CDP condition yielded a 2.95 Å structure in $P2_12_12_1$ (Supplementary Table 4). Initial refinement against the apo-NrdE monomer yielded significant unmodeled electron density at the S-, I-, and catalytic sites. Following several cycles of manual model building in Coot and refinement in Phenix, TTP-Mg$^{2+}$ was modeled into both S-sites in the ASU, ADP was modeled into the I-site density of both chains, and the C-terminal peptide was modeled into the catalytic sites of both chains. No omit or Polder omit density for a ligand was observed at the M-sites. The final model for the 2.95 Å NrdE S-dimer includes residues 8–269, 275–603, 610–685, and 694–700 in chain A; 7–237, 246–685, and 694–700 in chain B; two molecules of TTP; two molecules of ADP; and two Mg$^{2+}$.

To produce the crystal structures of disulfide-trapped (PDB: 6MT9) and X-ray-reduced (PDB: 6MVE) NrdE S-dimer, a highly redundant dataset was collected on a single crystal grown from the TTP/ATP/GDP condition to yield two structures: a 2.50 Å disulfide-trapped structure and a 2.55 Å X-ray-reduced structure from the second half of data collection (Supplementary Table 4). The 2.50 Å structure was refined in a similar manner as above. Clear electron density was observed at the M-site, S-site, I-site, and catalytic site. TTP-Mg$^{2+}$ was modeled into the S-site. As in the 2.95 Å structure, the electron density at the I-site was more consistent with ADP than with ATP. At the M-site, both ATP and GDP were initially considered. While the electron density is less resolved here than at the I- or S-sites, there is a notable lack of density for the N2 amine of a GDP, and there is also density for a γ-phosphate. Thus, ATP was modeled into the density rather than GDP. We note, however, that the binding site is likely to be able to bind GDP as well, and that the ATP may have displaced GDP when soaked in with cryoprotectant.

Density at the catalytic site was modeled as a peptide based on its shape and connectivity to catalytic-site residues C382 and C170. The C-terminus was then built via iterative manual residue placement in Coot and refinement in Phenix, using the disulfide bridges involving C695 and C698 as anchor points. Interestingly, these disulfides were not reduced by fresh TCEP that was introduced in the cryoprotectant. However, we were able to obtain a largely reduced structure by collecting a redundant diffraction dataset. Using these later frames, the structure was solved via rigid body refinement using the 2.50 Å oxidized structure as a starting model. The resulting 2.55 Å dataset (Supplementary Table 4) is at slightly lower resolution while retaining good merging statistics. The structure is virtually unchanged except that cysteines 382, 170, 695, and 698 were better modeled in the reduced form. However, we note that there is evidence for mixed disulfide occupancy in both the 2.50 and 2.55 Å structures (Supplementary Fig. 11). The final models for the 2.50 Å disulfide-trapped and 2.55 Å X-ray reduced NrdE structures include residues 6–238, 246–685, and 695–700 (2.50 Å) or 694–700 (2.55 Å); 1 molecule of TTP; 1 molecule of ADP; 1 molecule of ATP; 1 Mg$^{2+}$; and 194 (2.50 Å) or 133 (2.55 Å) water molecules.

**Reporting summary**. Further information on research design is available in the Nature Research Reporting Summary linked to this article.

## Data availability

Coordinates and structure factors for crystal structures have been deposited in the Protein Data Bank under the following accession codes: disulfide-trapped S-dimer (PDB 6MT9), X-ray-reduced S-dimer (PDB 6MVE), and S-dimer with empty M-sites (PDB 6MV9). EM structures and associated atomic models have been deposited in the Electron Microscopy Data Bank and the Protein Data Bank under the following accession codes: dATP-inhibited NrdEF filament (EMD-9272; PDB 6MW3) and dATP-inhibited NrdE filament (EMD-9293; PDB 6MYX). Source SAXS data underlying Figs. 2a, b, 3, and 5 are provided as a Source Data file. Other data are available from the corresponding author upon reasonable request.

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

## Acknowledgements

The authors thank Albert Kim and Dr. Mackenzie Parker for providing the NrdF samples used in this study. We are grateful to Dr. Parker for many helpful discussions; Dr. Brian Crane for critical reading of this manuscript; Max Watkins, Gabrielle Illava, Dr. Steve Meisburger, Dr. Jesse Hopkins, and Dr. Richard Gililan for assistance with X-ray data collection; Drew Gingerich and Adam Oken in the Chen lab for their assistance in NrdE filament cryo-EM data collection; and Sajid Fahumy in the Kaelber lab for assistance with NrdEF filament single-particle processing. Cryo-EM was performed at the Oregon Health & Science University Multiscale Microscopy Core and the Rutgers New Jersey Cryoelectron Microscopy and Tomography Core Facility. SAXS and crystallography were performed at the Cornell High Energy Synchrotron Source (CHESS), which is supported by the National Science Foundation under award DMR-1332208, using the Macromolecular Diffraction at CHESS (MacCHESS) facility, which is supported by award GM-103485 from the National Institute of General Medical Sciences, National Institutes of Health. This work was supported by NIH grants GM081393 (to J.S.) and GM124847 (to N.A.) and startup funds from Oregon Health & Science University (to J.Z.C.), Princeton University (to N.A.), and Cornell University (to N.A.).

## Author contributions

W.C.T. and N.A. designed the research and wrote the manuscript. W.C.T. purified protein and performed SAXS, crystallography, and EM, including data analysis, structure determination, and refinement of X-ray and EM models. F.P.B. purified protein and grew crystals used in this study, F.P.B and A.A.B. performed supporting experiments, and J.-P.B. performed crystallographic data processing and refinement of X-ray and EM models. J.S., J.T.K., and J.Z.C. contributed to experimental design and interpretation. J.T.K. and J.Z.C. performed EM data collection and single-particle reconstructions, and J.Z.C. developed software for helical reconstructions. N.A. collected X-ray data, performed research, and led the research. All authors edited the manuscript.

## Additional information

**Competing interests:** The authors declare no competing interests.

