## [Peer Review File · Nature Communications]

Reviewers' comments:

Reviewer #1 (Remarks to the Author):

Ribonucleotide reductases of the so-called Ib class are distinctive in lacking the so-called ATP cone, an element that contributes toward regulation of overall activity of the enzyme. Recently the authors of the present manuscript reported a Class Ib RNR (*Bacillus subtilis*) to be distinctive in containing a tightly bound dAMP molecule that modulates the control of enzyme activity by dATP. This suggests a novel form of activity regulation, which this study investigates at the structural level, using a combination of small-angle X-ray scattering, crystallography, and cryo-electron microscopy. The study describes in detail a novel form of activity regulation. Although the authors focus more on the "conformational gymnastics" of this novel regulation, they do point out that many pathogenic microorganisms possess only the class Ib RNR, which may identify this RNR as a drug target.

This is an extremely impressive paper. The authors describe six unique structures, which reveal the "conformational gymnastics" that control Ib RNRs via an "evolutionary convergent" form of allostery. The methods and interpretation of the data are clearly described, and the figures are a delight—both the in-text and supplementary figures.

Reviewer #2 (Remarks to the Author):

The manuscript by Thomas et al. describes a thorough investigation of the unusual structural mechanisms of allosteric activity regulation in the class Ib ribonucleotide reductase from *B. subtilis*, which until recently was assumed to lack activity regulation due to the absence of the ATP cone domain. The work, a combination of small-angle X-ray scattering, cryo-electron microscopy and X-ray crystallography, is very well-conceived and carefully carried out, and the conclusions are almost entirely well-justified. Thus the manuscript is certainly worthy of publication in *Nature Communications*. However there are a few issues that need to be cleared up.

1) The role of the so-called M-site discovered in the crystal structure of NrdE in complex with 5 mM ATP, 0.5 mM TTP and 1mM GDP is rather obscure and the relevant paragraph at the end of page 13 is speculative on how ATP came to be there. GDP is the substrate for specificity effector TTP. What would be the biological function of having the substrate bind here? For this particular effector-substrate combination one could accept that it promotes activity by dissociating I-dimers, but what about the other substrates? The more speculative parts of this section could be moved to the Discussion.

2) It is exciting and significant that the C-terminal segment of a class I RNR has finally been caught in the active site, but the exact significance of the observed conformation is unclear. The authors first describe it as an artefact of oxidation, particularly the disulphide bond between Cys695 and the initiator Cys382. This raises the question whether this could cause an artefactual conformation of the rest of the peptide. If Cys698 is, as written, "poised to initiate reduction" of the oxidised Cys170-Cys409 pair, then what is the role of the other conserved cysteine, Cys695? How does it get to its active conformation, which is presumably closer to Cys170-Cys409? This would require the peptide to be shifted towards the right in the view of Fig. 4g, but it seems locked in place by the interaction of the C-terminus. Do the authors believe that the observed Cys695-Cys382 disulphide may even have a role in the catalytic cycle? It's important to be clear about the logic, because if the peptide is artefactually displaced in the active site, this could also potentially affect the observed conformation of Tyr684, which is part of the "conformational gating" argument.

3) Why is the maximum observed Rg of the proposed filament induced by TTP (~110 Å) so much larger than that of the one induced by dATP (85 Å)? The value of 85 Å is reached at around 100 microM (which is presumably why SEC-SAXS was done at that concentration) but what do the authors think is happening above 100 microM?

Minor points

p. 3 line 7: "hypothesis-driven" is somewhat unnecessary, as that's the way all science should

be...

p. 7 Fig. 2c and p. 8 line 19: Why does the secondary peak characteristic of the NrdE filament disappear immediately upon addition of NrdF but the peak apparently characteristic of the NrdEF filament appear much more slowly?

p. 8 line 13: I was confused for some time by the statement that NrdF fills the "central core".

Surely there can't be a "core" in a single NrdE filament, but something more like a groove, as in DNA? The word "core" is also used on p. 9 line 9 to describe the alpha-helical part of NrdF, leading to further confusion. That could be clarified by calling it "core domain".

Cys382 is better described as "radical initiator Cys382" than "catalytic Cys382" as all three of the important cysteines are catalytic in some way.

p. 11, fourth last line: What was the "insight from SAXS"? The concentration of TTP?

Derek Logan

Reviewer #3 (Remarks to the Author):

Thomas et al. dissect the unique mechanism of activity regulation in the class Ib ribonucleotide reductase (RNR) from *Bacillus subtilis* by combining SAXS, cryo-EM and crystallographic data. From the SAXS data the authors deduce the existence of six distinct structures/states and their interconversions important for allostery. They characterize the oligomeric and global structural properties of the six states. Using this information, they define conditions to obtain cryo-EM structures of two novel inhibited oligomeric states, a filament of NrdE and a filament of NrdEF, and three crystal structures of the known canonical "S-dimer" of NrdE obtained under activating conditions. The cryo-EM structures suggest the structural basis of *Bacillus subtilis* RNR inhibition, i.e. using the partial NrdE ATP-cone to form filaments, in contrast to the well-studied class Ia RNRs, which possess full ATP-cones and form ring-shaped oligomers. The crystal structures of the NrdE S-dimer provide new insights into effects and modes of nucleotide recognition and into the role of the C-terminal residues in active site regeneration and conformational gating. In summary, the present work by Thomas et al. represents an important extension of their previous work from a partial model (PNAS, 115 (20), E4594-E4603) towards an overall model of activity regulation of the *Bacillus subtilis* class Ib RNR. However, there are several major points, as outlined below, that the authors have to address to warrant publication. Most importantly, there are discrepancies in the cryo-EM structure determination (major points 1 and 2), which have to be clarified to validate the cryo-EM structures.

Major points

1. Cryo-EM structure determination of the NrdE filament should be explained in more detail to allow validation, in particular as the authors use in this case unpublished software, the "PARTICLE" package.

a) To verify the validity of their approaches, the authors should provide data for application of PARTICLE on a well-studied helical specimen and describe the similarities and differences of individual steps to published approaches.

b) The individual steps of image processing, in particular the crucial estimation of initial helical parameters and the refinement of the final helical parameters, should be described and illustrated in more detail.

c) There appears to be an inconsistency between the lower resolution features in the individual class averages and the detailed features in the resulting composite helical segment (Supplementary Figure 3a). This should be clarified and better illustrated by showing the class averages with their corresponding assignments to the composite map and reprojections.

d) The authors mention a 2D classification step, but state that the initial selected 85,532 particles were also used for the final reconstruction, i.e. no particles were sorted out during the full processing pipeline. This is unusual and should be explained.

2. The cryo-EM map of the NrdEF filament looks very well-resolved for the NrdE part providing in some regions details like resolved sidechains and nucleotides. This is surprising considering a) the overall resolution of 4.7 Å and b) the rather low quality of the raw data shown in Supplementary Figure 3b).

a) The discrepancy between overall resolution and higher-resolution local features might be explained by the strong differences in local resolution between NrdE and NrdF parts and the use of

a local filter for final rendering. To clarify this the authors should provide a local resolution map (see point 1) and overall FSC curves computed separately for the NrdE helix and the densities assigned to NrdF.

b) The micrograph of the NrdE filament in Supplementary Figure 3a shows well-defined homogeneous filaments and the features of the corresponding map correlate well with the overall resolution of 4.8 Å. The micrograph of the NrdEF filament sample, in contrast, looks very heterogeneous showing elongated tubes with variations in diameter, aggregates and smaller, broken assemblies. The authors should provide more details on all steps of image processing to explain the discrepancy between the lower-quality raw data and the highly resolved map features of the NrdEF filaments. They should provide a gallery of corresponding single particles, 2d class averages, surface representations of the ab initio 3d map and reprojections of the 3D map. Furthermore, the authors should clarify, if they have used any models during image processing, e.g. in particle selection or ab initio structure determination. According to Methods, the data have been processed in cryo-Sparcs w/o symmetry, but the density shows a cutout from a continuous helix - how can this be explained?

c) No classification of the NrdEF filament data is mentioned, the number of initial particles equals the number of final particles. Considering the heterogeneity in the raw data, this is very unusual and should be explained.

3. The very weak density in the NrdEF filament assigned by the authors to the NrdF core is not enough defined to allow for fitting or any interpretation (Figure 2e, Supplementary Figure 4b). The authors should either remove the corresponding parts or substantially improve the density, e.g. by focused classification. The "test for potential bias in map refinement" suggests that there is no other preferred conformation of NrdF (p. 9, l.14-17 and Supplementary Figure 3b), but also the test cannot exclude the sampling of NrdF conformations compatible with radical transfer.

4. The authors propose an overall model of activity regulation, as shown in Figure 7. However, there is no data on the depicted reversible transition between inhibited NrdEF filaments and active NrdEF tetramers. The authors should provide evidence for the transition.

5. Why is there no detailed SAXS analysis on the NrdEF filament, as provided for the NrdE filament? Considering the heterogeneous cryo-EM raw data data of the NrdEF filament sample (Supplementary Figure 3b), such an analysis would be helpful to corroborate the importance of the NrdEF filament.

6. In the cell, both NrdE and NrdF are present. Therefore, a direct transition from NrdE and NrdF dimers or NrdEF tetramers to NrdEF filaments seems more plausible than the described detour via formation of NrdE filaments and subsequent replacement of just one NrdE helix by NrdF. Did the authors test conditions for the former pathway? Is there evidence that the route via NrdE filaments is the physiological relevant one? How can it be explained that just one NrdE helix of the double-helical NrdE filament should be segregated and replaced by NrdF dimers?

Minor points

7. In the cryo-EM part, several standard data for validation are missing. In particular, FSC curves from independent half-maps, map vs. model FSC curves and local resolution maps (e.g. computed with ResMap) should be added.

8. Figure 2d-h and Supplementary Figure 4: The absolute threshold values for the cryo-EM maps should be replaced by the statistically more meaningful σ levels (standard deviations from the mean).

9. Figure 2g,h: It should be clarified, how the difference densities were computed.

10. Supplementary Table 2 and Methods: The total electron dose used for NrdEF cryo-EM data acquisition should be clarified: 5-11 electrons per Å² appear very low and might rather correspond to the electron flux per second. For NrdE filament processing, the table states that no symmetry was used, whereas according to Methods helical symmetry was applied; this should be clarified.

11. Abstract, l.9: "Conformational changes" appears more appropriate than "conformational gymnastics".

12. P. 5, l. 9: The term "mathematically" is misleading in this context and should be omitted.

Please find enclosed our revised manuscript, “Convergent Allostery in Ribonucleotide Reductase.”

We thank the reviewers for their overall positive comments and thoughtful questions. We have addressed all questions and believe that they have led to a better manuscript. Our revised manuscript contains a new SI table and 3 additional SI figures. Reviewer #1 found no issues. Changes to the manuscript in response to Reviewers #2 and 3 are highlighted in cyan and yellow, respectively, as detailed below.

Reviewer #1:

... This is an extremely impressive paper. The authors describe six unique structures, which reveal the “conformational gymnastics” that control Ib RNRs via an “evolutionary convergent” form of allostery. The methods and interpretation of the data are clearly described, and the figures are a delight—both the in-text and supplementary figures.

We thank the reviewer for the enthusiastic response.

Reviewer #2:

... The work, a combination of small-angle X-ray scattering, cryo-electron microscopy and X-ray crystallography, is very well-conceived and carefully carried out, and the conclusions are almost entirely well-justified. Thus the manuscript is certainly worthy of publication in Nature Communications. However there are a few issues that need to be cleared up.

We thank the reviewer for raising important questions. We have revised the text accordingly, and we have expanded Supplementary Figure 8 into two separate figures (now **Supplementary Figures 10 and 12**).

1) The role of the so-called M-site discovered in the crystal structure of NrdE in complex with 5 mM ATP, 0.5 mM TTP and 1mM GDP is rather obscure and the relevant paragraph at the end of page 13 is speculative on how ATP came to be there. GDP is the substrate for specificity effector TTP. What would be the biological function of having the substrate bind here? For this particular effector-substrate combination one could accept that it promotes activity by dissociating I-dimers, but what about the other substrates? The more speculative parts of this section could be moved to the Discussion.

We agree that an allosteric role for GDP would be unexpected considering it is a substrate. We had two reasons to discuss its potential role at the M-site, which are now explicitly stated on p. 13-14 and 19. First, we found that of 27 diffraction-quality crystals that were grown with various combinations of ATP, TTP, GDP, and CDP, electron density at the M-site was observed *only* when the crystallization condition contained GDP. Secondly, an analytical ultracentrifugation study by Mackenzie Parker and JoAnne Stubbe showed that *B. subtilis* NrdE becomes monomeric in the presence of GDP [Parker, M. J. Doctoral dissertation (2017): <http://hdl.handle.net/1721.1/109681>]. Because ATP was bound at the M-site of our crystal, we could not neglect its possible role in our allosteric model. However, we did not want to misleadingly suggest that ATP must be the sole physiological effector at the M-site.

Regarding biological significance, it is known that guanine nucleotides have an unusual role in *B. subtilis*, a model organism for sporulation [Lopez, et al. *Biochim Biophys Acta*. (1979) 587:238-252]. Sporulation activates the conversion of GDP/GTP into (p)ppGpp alarmones. Indeed, a possible connection has been proposed between stress response and RNR inhibition in *B. subtilis* [Parker Doctoral dissertation (2017)]. Such a mechanism would certainly be interesting, and we are actively investigating this for a future study.

2) It is exciting and significant that the C-terminal segment of a class I RNR has finally been caught in the active site, but the exact significance of the observed conformation is unclear. The authors first describe it as an artefact of oxidation, particularly the disulphide bond between Cys695 and the initiator Cys382. This raises the question whether this could cause an artefactual conformation of the rest of the peptide. If Cys698 is, as written, "poised to initiate reduction" of the oxidised Cys170-Cys409 pair, then what is the role of the other conserved cysteine, Cys695? How does it get to its active conformation, which is presumably closer to Cys170-Cys409? This would require the peptide to be shifted towards the right in the view of Fig. 4g, but it seems locked in place by the interaction of the C-terminus. Do the authors believe that the observed Cys695-Cys382 disulphide may even have a role in the catalytic cycle? It's important to be clear about the logic, because if the peptide is artefactually displaced in the active site, this could also potentially affect the observed conformation of Tyr684, which is part of the "conformational gating" argument.

We thank the reviewer for appreciating the significance of capturing the α C-terminus while raising important questions. We have revised the text on p. 14-15 and added a new figure (**Supplementary Fig. 10**) to clarify the following points:

- Although both disulfides are likely a result of crystal oxidation, the catalytic site is minimally perturbed by the bound C-terminus. Our crystal structures are nearly superimposable with previous structures of *B. subtilis* NrdE with a reduced catalytic site (pdb: 6CGN, C α rmsd = 0.8 Å) as well as *S. typhimurium* NrdE with an oxidized catalytic site (pdb: 1PEO, C α rmsd = 1.0 Å). Numerous tail-binding interactions are observed in our structures, and all of the residues that interact with the disulfide-trapped C-terminus in our structure are conserved in *S. typhimurium*.
- We believe that the C170-C698 disulfide is captured in a physiologically relevant conformation. As the reviewer is aware, re-reduction of the catalytic-site cysteines is expected to require two sequential S_N2 nucleophilic substitutions. Alignment of the *S. typhimurium* NrdE structure (**Supplementary Fig. 10c**, dark gray) with the C-terminus from our X-ray-reduced structure (**Supplementary Fig. 10c**, purple sticks) places C698 within 3 Å of the oxidized C178-C415 pair in the active site (equivalent to C170-C409 in *B. subtilis* numbering). These three cysteines form a triad that is optimal for disulfide exchange, with C698 well positioned to attack C178 (C170 in *Bs*). We thus believe that the C-terminal region of the tail containing C698, C170, and C409 is bound in a physiologically relevant conformation necessary for the first step of catalytic site re-reduction. Perhaps as a result, we observe an oxidized C698-C170 pair in our disulfide-trapped structure. We note that the very C-terminal region of the tail is also where most of the conserved tail-binding interactions are concentrated.
- We believe that C695 is captured in an artifactual conformation. In the second step of re-reduction, C695 is expected to attack the C698-C170 disulfide. Alignment of our structure with the disulfide-trapped C-terminus (**Supplementary Fig. 10d**, white and purple sticks) with C382 in its reduced conformation from our X-ray-reduced structure (**Supplementary Fig. 10d**, blue sticks) shows that C695 is too distant from C698 (5.6 Å) to be compatible with disulfide exchange with the C698-C170 disulfide.
- There is one non-conserved interaction that we resolve in our X-ray-reduced structure, which we believe is locking C695 in place. In this structure, the residue adjacent to C695 (S694) forms an H-bond with S246, a catalytic site residue that is *not* conserved. We thus speculate that this non-conserved interaction restricted C695 from moving closer to C698-C170 and instead led to its artifactual oxidation with the nearby C382.
- Regardless, it is important to note that moving C695 closer to C698-C170 would cause the C-terminus to pull the β -strand containing the Y683-Y684 dyad even more. Hence, our observed C695 conformation does not affect our assertion that tail binding leads to the unstacking of the Tyr684.

3) Why is the maximum observed R_g of the proposed filament induced by TTP (~110 Å) so much larger than that of the one induced by dATP (85 Å)? The value of 85 Å is reached at around 100 microM (which is presumably why SEC-SAXS was done at that concentration) but what do the authors think is happening above 100 microM?

The reviewer is correct that the final R_g reached in the holo-NrdE titration of dATP is lower than that of the TTP titration. This is simply because we did not saturate the dATP titration. The two titrations were performed on separate synchrotron trips, and thus the low- q limit (set by the beamstop) differs for the two datasets. We terminated the dATP titration before reaching a higher R_g because the low- q limit did not justify performing Guinier analysis. We note that R_g as a measure of saturation becomes increasingly less meaningful for heterogeneous filamentous samples. A more meaningful comparison is obtained by SEC-SAXS analyses (**Supplementary Fig. 2b-c**), which show that the dATP and TTP filaments are remarkably similar. We now explicitly state in the **caption** that the R_g 's determined by SEC-SAXS are ~100 Å for both filaments.

Minor points

p. 3 line 7: "hypothesis-driven" is somewhat unnecessary, as that's the way all science should be...

We have rephrased this sentence on p. 5 to clarify our intended meaning.

p. 7 Fig. 2c and p. 8 line 19: Why does the secondary peak characteristic of the NrdE filament disappear immediately upon addition of NrdF but the peak apparently characteristic of the NrdEF filament appear much more slowly?

When a subunit titration is performed past stoichiometric amounts, the resulting structural changes always involve at least three species, and thus, the disappearance of a structural feature will not necessarily be concomitant with the appearance of another as in a two-state transition. In the case of this NrdEF titration, it can be shown by singular value decomposition (SVD) that there are in fact four species (**Supplementary Fig. 4d**). Below sub-stoichiometric amounts of NrdF, there is a rapid loss of the NrdE double helix and formation of the NrdEF filament. This is followed by the buildup of excess NrdF, which gives rise to a mid- q shoulder in the Kratky plot shown in Fig. 2c. A unique feature of this subunit titration is the presence of a fourth species that is associated with a small spike at $q = 0.092 \text{ \AA}^{-1}$. A peak at this position is consistent with the association of NrdF to the NrdE filament, while the sharpness of this reproducible feature is reminiscent of fiber diffraction. As we will discuss in detail in response to Reviewer #3, this feature arises from the tendency of the NrdEF filaments to associate into crystalline-like bundles.

Thus, the complex interconversion of NrdE double-helices, single NrdEF filaments, free NrdF, and NrdEF filament bundles is reflected in our SAXS profiles as a loss of the double-helix feature at $q = 0.055 \text{ \AA}^{-1}$ prior to the formation of the feature at $q = 0.092 \text{ \AA}^{-1}$. We now explicitly state this in the Fig. 2c caption (p. 3), and the SVD analysis and an electron micrograph of the NrdEF filament bundle are shown in **Supplementary Figs. 4d and 4b**, respectively.

p. 8 line 13: I was confused for some time by the statement that NrdF fills the "central core". Surely there can't be a "core" in a single NrdE filament, but something more like a groove, as in DNA? The word "core" is also used on p. 9 line 9 to describe the alpha-helical part of NrdF, leading to further confusion. That could be clarified by calling it "core domain".

This is an excellent point, and we thank the reviewer for bringing it to our attention. The filament "core" is now referred to as the filament "interior" throughout the paper.

Cys382 is better described as "radical initiator Cys382" than "catalytic Cys382" as all three of the important cysteines are catalytic in some way.

We have revised accordingly. See p. 2 and 14 (bottom of page).

p. 11, fourth last line: What was the "insight from SAXS"? The concentration of TTP?

Yes, but more specifically, SAXS results allowed us to systematically improve the crystallization condition for the S-dimer. We have added a reference to the SI Methods on p. 12 to clarify what we meant by "insight".

Reviewer #3:

... In summary, the present work by Thomas et al. represents an important extension of their previous work from a partial model (PNAS, 115 (20), E4594-E4603) towards an overall model of activity regulation of the Bacillus subtilis class Ib RNR. However, there are several major points, as outlined below, that the authors have to address to warrant publication. Most importantly, there are discrepancies in the cryo-EM structure determination (major points 1 and 2), which have to be clarified to validate the cryo-EM structures.

We thank the reviewer for raising compelling points. We agree that detailed EM methods and validation analyses should be in the manuscript. We have revised the text accordingly and significantly expanded the SI Methods. Supplementary Figure 3 has been expanded into three figures (now **Supplementary Figure 3-5**). Detailed responses are below.

Major points

1. Cryo-EM structure determination of the NrdE filament should be explained in more detail to allow validation, in particular as the authors use in this case unpublished software, the "PARTICLE" package.

a) To verify the validity of their approaches, the authors should provide data for application of PARTICLE on a well-studied helical specimen and describe the similarities and differences of individual steps to published approaches.

The PARTICLE package builds upon previously published work [Barthelme, Chen, et al. *PNAS* (2014) 111: E1687–E1694] and has been benchmarked in the 2015 CTF Estimation Challenge [Marabini, et al. *J. Struct. Biol.* (2015) 190:348–359]. The helical reconstruction function in PARTICLE was developed and validated on TMV datasets from both the Chen lab and the public database EMPIAR. A separate manuscript detailing the image processing functions in the PARTICLE package and comparisons with existing methods is currently in preparation. In the specific case of NrdE filament reconstruction (85,532 particles, images at 1.51 Å/pixel), the result is directly validated by its agreement with the corresponding density features in the NrdEF density map (126,224 particles, images at 1.05 Å/pixel), which has been processed independently in cryoSPARC v2 and Relion. Importantly, the NrdE filament model also agrees with the experimental scattering obtained by SAXS.

b) The individual steps of image processing, in particular the crucial estimation of initial helical parameters and the refinement of the final helical parameters, should be described and illustrated in more detail.

The SI Methods have been expanded on p. 28-30 to better describe the image processing procedure in PARTICLE. We now dedicate **Supplementary Figure 3** entirely to better illustrate this process.

c) There appears to be an inconsistency between the lower resolution features in the individual class averages and the detailed features in the resulting composite helical segment (Supplementary Figure 3a).

This should be clarified and better illustrated by showing the class averages with their corresponding assignments to the composite map and reprojections.

The class averages in the original figure were 2x-binned in the display and therefore did not offer the same detail as in the composite filament. In the updated **Supplementary Figure 3b**, we have replaced this montage. In the new montage, class averages contributing to the composite filament have been placed on both sides and vertically aligned to the respective features in the composite filament image. Additionally, a reprojection of the reconstructed density map (low-pass filtered down to 15 Å) is juxtaposed beneath the composite filament from class averages to illustrate class-map agreement.

d) The authors mention a 2D classification step, but state that the initial selected 85,532 particles were also used for the final reconstruction, i.e. no particles were sorted out during the full processing pipeline. This is unusual and should be explained.

All filaments processed in the 3D reconstruction have been manually annotated from data micrographs, during which special care has been applied to identifying “good” usable particles. In addition, the contribution of each particle frame is weighted into the 3D reconstruction in PARTICLE, with the weighting function being the phase-residual score in particle alignment (now stated on p. 30). As a result, “bad” particles are also included in the 3D reconstruction, though at much lower weight in contribution.

2. The cryo-EM map of the NrdEF filament looks very well-resolved for the NrdE part providing in some regions details like resolved sidechains and nucleotides. This is surprising considering a) the overall resolution of 4.7 Å and b) the rather low quality of the raw data shown in Supplementary Figure 3b).

a) The discrepancy between overall resolution and higher-resolution local features might be explained by the strong differences in local resolution between NrdE and NrdF parts and the use of a local filter for final rendering. To clarify this the authors should provide a local resolution map (see point 1) and overall FSC curves computed separately for the NrdE helix and the densities assigned to NrdF.

We appreciate that the reviewer finds our NrdEF map “very well-resolved.” We agree that features beyond 4.7 Å are readily discernible in the reconstruction. The reason for this is as the reviewer suspected: strong differences in local resolution between NrdE and NrdF. We have revised the text on p. 8 and now provide a local resolution map in **Supplementary Fig. 5b** and FSC curves in **Supplementary Fig. 5d** to help clarify this point. We took a conservative approach by reporting the global resolution of the map, but we agree that the overall map resolution is reflective of the presence of high-resolution features in NrdE and the absence of such features in NrdF.

b) The micrograph of the NrdE filament in Supplementary Figure 3a shows well-defined homogeneous filaments and the features of the corresponding map correlate well with the overall resolution of 4.8 Å. The micrograph of the NrdEF filament sample, in contrast, looks very heterogeneous showing elongated tubes with variations in diameter, aggregates and smaller, broken assemblies. The authors should provide more details on all steps of image processing to explain the discrepancy between the lower-quality raw data and the highly resolved map features of the NrdEF filaments. They should provide a gallery of corresponding single particles, 2d class averages, surface representations of the ab initio 3d map and reprojections of the 3D map.

We thank the reviewer for bringing the quality of our NrdEF micrograph to our attention. We have replaced this image with a more representative and informative micrograph (shown in **Supplementary Fig. 4a**). We assessed the raw data quality by examining the resolution at which Thon rings could be fit in the radially-averaged power spectrum. By this metric, the quality of the data is quite compatible with the final resolution, despite the apparent heterogeneity. The CTF maximum resolution as estimated by Gctf was < 4Å for 44,920 particles.

As mentioned in response to Reviewer 2, we also found by SAXS that the NrDEF filament tends to form ordered assemblies that give rise to a small but sharp scattering feature. We therefore also present an image in **Supplementary Figure 4b** depicting the crystalline bundles that were infrequently but reproducibly observed in micrographs of NrDEF filaments. Thus, although NrDEF filaments are more heterogeneous in length and more flexible than the NrDE double-helix, they are in fact ordered enough to self-assemble into crystalline rafts.

We also agree with the reviewer that more detailed methods should be included in the manuscript. To address this, we have expanded the SI Methods on p. 31-32 and added a gallery of representative 2D class averages (and corresponding reprojections) to **Supplementary Fig. 4c**. In addition, we provide below a gallery of all class averages. Subsetting of these particles by excluding “bad” classes, however, did not quantitatively or qualitatively improve the final reconstructed map. We describe this in further detail in response to part c.

Furthermore, the authors should clarify, if they have used any models during image processing, e.g. in particle selection or ab initio structure determination. According to Methods, the data have been processed in cryo-Sparcs w/o symmetry, but the density shows a cutout from a continuous helix - how can this be explained?

We can confirm that we did not use models during particle selection or *ab initio* structure determination. No models were used to go from raw data to the final deposited map. Models were used when we at-

tempted to bias the map towards various proposed $\alpha_2\beta_2$ configurations; in this experiment we found that the data was not consistent with these configurations. No results from this biasing experiment were propagated forward to other refinements, so no RNR models from outside our cryoEM dataset influenced the final reconstruction. We agree that the methods were not adequately described in the original manuscript and have clarified these points on p. 31-32.

The density we present in the manuscript shows a cutout from a continuous helix because the box size we selected for reconstruction was large enough to contain multiple repeats of NrdEF. The helical symmetry emerged naturally in refinement and was not imposed. When refining helical specimens in software designed for isolated single particles, one should use enough padding to avoid edge artifacts. We thus began with a large box size (~500 Å) when extracting data from micrographs. This box included several helical repeats: 9-12 copies of the NrdEF ASU. Then, to avoid edge effects, we extracted the central region (~250 Å) for rendering and analysis. Because of radial and/or edge effects, the central repeats are well-resolved and repeats further away become blurred, as is shown to the right.

We have deposited the unclipped, unfiltered half-maps in the EMDDB as well as the clipped, filtered, masked final map that was used for making figures. As can be seen from this map and figures in the manuscript, the helical symmetry naturally emerges in the C1-refined map. We have now expanded the SI Methods on p. 31-32 to explicitly describe the boxing and clipping steps.

c) No classification of the NrdEF filament data is mentioned, the number of initial particles equals the number of final particles. Considering the heterogeneity in the raw data, this is very unusual and should be explained.

In addition to performing 3D reconstruction of the full dataset, we also attempted classification and performed 3D reconstruction with a subset pruned from the best classes and representing 58% of the data. While reconstruction from the full dataset reached a resolution of 4.65 Å (as measured by cryoSPARC v2), reconstruction of the subset reached a resolution of 4.72 Å. Visual examination did not reveal any significant disagreement between the maps, though the map refined with the entire dataset seemed more crisp by eye. The FSC between the two maps without any masking or filtration crosses 0.5 at 5.1 Å, while the FSC between the two maps after masking and filtration crosses 0.5 at 4.1 Å. This indicates that these maps are generally the same to the interpreted resolution, and hence does not change the conclusions that we make. We suspect that classification did not help because all particles were boxed manually, a laborious step that is typically automated these days (with concomitant loss of accuracy). We have revised the SI Methods on p. 31 to describe the particle boxing process and clarify that the classification experiment was performed but did not improve resolution.

3. The very weak density in the NrdEF filament assigned by the authors to the NrdF core is not enough defined to allow for fitting or any interpretation (Figure 2e, Supplementary Figure 4b). The authors should either remove the corresponding parts or substantially improve the density, e.g. by focused classification. The “test for potential bias in map refinement” suggests that there is no other preferred conformation of NrdF (p. 9, l.14-17 and Supplementary Figure 3b), but also the test cannot exclude the sampling of NrdF conformations compatible with radical transfer.

We have removed the fitting of β_2 from **Figure 2e**. As noted by the reviewer, this density is not well-defined enough to indicate the orientation of the NrdF core. Attempts to substantially improve the density have thus far been unsuccessful but do warrant investigation in future studies. However, regardless of the exact orientation of the NrdF core, the density is of sufficient strength to suggest that it is on average placed far from NrdE, forming a central column within the helical interior of the filament. The density has the appropriate size and shape to fit one dimer of NrdF core, and the “test for potential bias in map refinement”, now better described on p. **31-32**, suggests that at a minimum, its center of mass is correctly placed—this conclusion could be drawn even if the map resolution were 40 Å. We have revised the text on p. **9-10** to remove any conclusions that require a specific orientation of NrdF core dimer. We also agree with the reviewer that we cannot interpret the density to mean that the NrdF core adopts a single conformation in which radical transfer is prevented. We therefore place greater significance on the way in which NrdF is confined within the helical interior and how this arrangement can hinder the sequence of motions needed for multiple turnovers. We have revised the text on p. **9-10** and **18-19** and believe that we now present a much stronger and more nuanced interpretation.

4. The authors propose an overall model of activity regulation, as shown in Figure 7. However, there is no data on the depicted reversible transition between inhibited NrdEF filaments and active NrdEF tetramers. The authors should provide evidence for the transition.

In Figure 3b, we show direct evidence that addition of ATP leads to the dissociation of the dATP-inhibited NrdE filaments into S-dimers. As shown by cryo-EM, the ASU of the dATP-inhibited NrdEF filaments is an $\alpha_2\beta_2$ composed of a NrdE S-dimer and a NrdF dimer, where each ASU is linked via I-dimer interactions. We thus think it is reasonable to propose that, like in NrdE filaments, addition of ATP will disrupt the I-dimer interfaces within the NrdEF filament and lead to its dissociation into NrdEF tetramers.

In the original manuscript, we described that we performed SEC-SAXS on NrdEF with various nucleotide mixtures but did not explicitly say that without both ATP and a specificity effector present, the oligomeric mixture often included species that are larger than $\alpha_2\beta_2$. We now clarify this point on p. **16**. The fact that we obtain a nearly pure sample of $\alpha_2\beta_2$ with ATP and a specificity effector is actually quite remarkable. It has not been directly shown, for example, that addition of ATP to the dATP-inhibited $\alpha_4\beta_4$ of *E. coli* class Ia RNR leads to a pure sample of $\alpha_2\beta_2$. In summary, we believe that we have sufficient evidence to propose in our model that ATP is able to break down extended structures of NrdEF.

5. Why is there no detailed SAXS analysis on the NrdEF filament, as provided for the NrdE filament? Considering the heterogeneous cryo-EM raw data data of the NrdEF filament sample (Supplementary Figure 3b), such an analysis would be helpful to corroborate the importance of the NrdEF filament.

To perform detailed structural modeling (e.g. comparisons with theoretical profiles), a high-quality scattering profile of a conformationally pure state is needed. Unfortunately, this measurement was not possible for the NrdEF filament due to its tendency to form bundles (**Supplementary Figure 4b**) and precipitate at the high loading concentrations needed for SEC-SAXS (this is to account for the ~10-fold dilution that occurs on the column; the final eluting concentration is designed to be in the low μM range). However, in response to Reviewer 2, we now provide a detailed SAXS analysis of the formation of the NrdEF filament in **Supplementary Figure 4d**. As described above, we also now provide a better image of the NrdEF cryo-EM raw data in **Supplementary Figure 4a** that shows that the NrdEF filament is the predominant species.

6. In the cell, both NrdE and NrdF are present. Therefore, a direct transition from NrdE and NrdF dimers or NrdEF tetramers to NrdEF filaments seems more plausible than the described detour via formation of NrdE filaments and subsequent replacement of just one NrdE helix by NrdF. Did the authors test condi-

tions for the former pathway? Is there evidence that the route via NrdE filaments is the physiological relevant one? How can it be explained that just one NrdE helix of the double-helical NrdE filament should be segregated and replaced by NrdF dimers?

We agree that in the cell, NrdE and NrdF co-exist with a mixture of nucleotides and that they may spontaneously form a NrdEF filament when dATP binds. We have performed control experiments to demonstrate that order of addition *does not* matter for NrdEF filament formation. We therefore agree that each type of allosteric site does not need to saturate with a ligand before another allosteric site undergoes the next binding event. However, we should note that even a “direct” pathway from NrdE and NrdF still consists of elementary steps, such as those depicted in our model in Figure 6. If we consider kinetics, it is certainly possible that multiple steps may occur with no significant buildup of intermediates.

In response to the reviewer’s last point, we do not think that just one NrdE helix of the double-helical NrdE filament would be segregated and replaced by NrdF. As we describe in the text, we believe that the NrdF C-termini compete for the NrdE double-helical interface. Thus, even substoichiometric amounts of NrdF can lead to the dissociation of the NrdE double-helix into single helices. Once dissociated into single helices, they are identical, and NrdF dimers should have no preference as to which strand they bind. As a result, NrdF will bind stoichiometrically to each NrdE helix, as supported by our titration study in **Figure 2c** and **Supplementary Figure 4d**.

7. In the cryo-EM part, several standard data for validation are missing. In particular, FSC curves from independent half-maps, map vs. model FSC curves and local resolution maps (e.g. computed with ResMap) should be added.

We agree and now include a new figure (**Supplementary Fig. 5**) to address this point.

Minor changes

8. Figure 2d-h and Supplementary Figure 4: The absolute threshold values for the cryo-EM maps should be replaced by the statistically more meaningful σ levels (standard deviations from the mean).

We reported absolute thresholds as they were required by the EMDB and there is no consensus in the field as to how σ levels should be reported. However, we have revised the text on p. 33 and now provide estimates in **Supplementary Table 4**. We note that standard deviations from the mean value are statistically meaningful in the unmasked, unfiltered half-maps but not in the locally filtered final map after solvent has been masked out. We provide estimates of both when possible. Perhaps the most pertinent information for the reviewer is the σ level for the NrdF density (2.0).

9. Figure 2g,h: It should be clarified, how the difference densities were computed.

Difference densities were computed with the phenix.real_space_diff_map routine in Phenix. This function calculates a difference map between a PDB model and an EM map in question after generating a simulated map of the PDB model at a specified resolution. In Figure 2g-h, we used a model of an α_4 (two S-dimers associating at the I-dimer interface) with ligands removed, set at a resolution of 4.0 Å. We have expanded the description on p. 33.

10. Supplementary Table 2 and Methods: The total electron dose used for NrdEF cryo-EM data acquisition should be clarified: 5-11 electrons per Å² appear very low and might rather correspond to the electron flux per second. For NrdE filament processing, the table states that no symmetry was used, whereas according to Methods helical symmetry was applied; this should be clarified.

We thank the reviewer for catching these errors! $5 \cdot 10^{-11} \text{ e}^-/\text{\AA}^2$ did refer to the dose per second rather than the total dose, and helical symmetry was applied for the NrdE filament processing. **Supplementary Table 2** has been corrected.

11. Abstract, l.9: “Conformational changes” appears more appropriate than “conformational gymnastics.

In the Introduction, we describe the fact that catalysis involves multiple sequential steps: 1) radical transfer (which requires a compact $\alpha_2\beta_2$ to form and part of the NrdF tail to fold into the subunit interface), 2) nucleotide reduction (and the implied reverse radical transfer back to NrdF), and 3) re-reduction of the catalytic site (which requires the $\alpha_2\beta_2$ interface to open and allow for the NrdF tail to exit and for the NrdE tail to enter). We think “conformational gymnastics” more precisely describes both the complexity and the sequential type of motions involved.

12. P. 5, l. 9: The term “mathematically” is misleading in this context and should be omitted.

Evolving factor analysis (EFA) is a mathematical technique that relies on iterative singular value decomposition (SVD) to identify changes in the rank of the data matrix. A common misconception is that size exclusion chromatography is sufficient to separate oligomeric mixtures. However, this is only true if the interconversion rate is slower than the already slow time scales of size exclusion. We developed EFA as a way to determine a rotation matrix that maps the singular vector basis set onto a physical basis set. This, in turn, yields scattering profiles that represent “unmixed” individual species. We feel that it would be more misleading to omit the term “mathematically”. Our method is described and cited in the SI Methods on p. **27**.

Reviewers' comments:

Reviewer #2 (Remarks to the Author):

The authors have responded to all my initial queries and those of the other reviewers, which has resulted in a substantially clearer and improved manuscript. I thank the authors particularly for their clarification of their ideas on the role of the C-terminal cysteines.

Derek Logan

Reviewer #3 (Remarks to the Author):

The revised manuscript addresses most of my points. Importantly, the extended cryo-EM methods now allows for much better validation. There is only one point left to warrant publication: The data for the NrdF filament strongly suggest an actual resolution of 6Å, rather than the claimed 4.8Å resolution (based on FSC curve of half-maps in Suppl. Fig. 5c). The shoulder in the FSC curve of half-maps (at 6Å resolution and higher) indicates overfitting and, accordingly, the model-map curve also rather indicates a resolution of about 6Å. Also the features in the cryo-EM map correspond to 6Å (alpha helices generally resolved), but not 5Å (the beta-strands are not resolved). Consequently, the claimed resolution of 4.8Å is a clear overestimation and the value in the main text should be changed to 6Å. This will not change any of the authors' conclusions, but make their work technically more sound.

Please find enclosed our revised manuscript, “Convergent Allostery in Ribonucleotide Reductase.”

Reviewer #2

The authors have responded to all my initial queries and those of the other reviewers, which has resulted in a substantially clearer and improved manuscript. I thank the authors particularly for their clarification of their ideas on the role of the C-terminal cysteines.

Derek Logan

We thank Derek Logan for his helpful and encouraging comments.

Reviewer #3:

The revised manuscript addresses most of my points. Importantly, the extended cryo-EM methods now allows for much better validation. There is only one point left to warrant publication: The data for the NrdF filament strongly suggest an actual resolution of 6Å, rather than the claimed 4.8Å resolution (based on FSC curve of half-maps in Suppl. Fig. 5c). The shoulder in the FSC curve of half-maps (at 6Å resolution and higher) indicates overfitting and, accordingly, the model-map curve also rather indicates a resolution of about 6Å. Also the features in the cryo-EM map correspond to 6Å (alpha helices generally resolved), but not 5Å (the beta-strands are not resolved). Consequently, the claimed resolution of 4.8Å is a clear overestimation and the value in the main text should be changed to 6Å. This will not change any of the authors' conclusions, but make their work technically more sound.

The reviewer asks for a reasonable and simple change regarding the NrdE (not NrdF) filament. We agree with the reviewer and have changed the resolution of the NrdE filament map from 4.8 to 6 Å, everywhere in the text (highlighted in yellow).

Manuscript file:

- p. 7, Fig. 2d caption
- p. 8
- p. 30, in EM Methods

SI Tables and Figures file:

- p. 3, Supplementary Table 2
- p. 10, Supplementary Fig. 5c caption